# A quantized mechanism for activation of pannexin channels

Yu-Hsin Chiu[1], Xueyao Jin[2,*], Christopher B. Medina[3,4,*], Susan A. Leonhardt[2], Volker Kiessling[2,5], Brad C. Bennett[2], Shaofang Shu[1], Lukas K. Tamm[2,5], Mark Yeager[2,5], Kodi S. Ravichandran[3,4] & Douglas A. Bayliss[1,5]

Pannexin 1 (PANX1) subunits form oligomeric plasma membrane channels that mediate nucleotide release for purinergic signalling, which is involved in diverse physiological processes such as apoptosis, inflammation, blood pressure regulation, and cancer progression and metastasis. Here we explore the mechanistic basis for PANX1 activation by using wild type and engineered concatemeric channels. We find that PANX1 activation involves sequential stepwise sojourns through multiple discrete open states, each with unique channel gating and conductance properties that reflect contributions of the individual subunits of the hexamer. Progressive PANX1 channel opening is directly linked to permeation of ions and large molecules (ATP and fluorescent dyes) and occurs during both irreversible (caspase cleavage-mediated) and reversible (α1 adrenoceptor-mediated) forms of channel activation. This unique, quantized activation process enables fine tuning of PANX1 channel activity and may be a generalized regulatory mechanism for other related multimeric channels.

[1] Department of Pharmacology, University of Virginia School of Medicine, Charlottesville, Virginia 22908, USA. [2] Department of Molecular Physiology and Biological Physics, University of Virginia School of Medicine, Charlottesville, Virginia 22908, USA. [3] Department of Microbiology, Immunology and Cancer Biology, University of Virginia School of Medicine, Charlottesville, Virginia 22908, USA. [4] Center for Cell Clearance, University of Virginia School of Medicine, Charlottesville, Virginia 22908, USA. [5] Center for Membrane and Cell Physiology, University of Virginia School of Medicine, Charlottesville, Virginia 22908, USA. * These authors contributed equally to this work. Correspondence and requests for materials should be addressed to D.A.B. (email: bayliss@virginia.edu).

Pannexin 1 (PANX1) is a widely expressed oligomeric membrane channel, in which each subunit has four putative transmembrane domains. PANX1 channels are topologically related to connexins and invertebrate innexins, and to the recently identified CALHM1 and SWELL1 (LRRC8) channels[1–3]. PANX1 channels are activated by diverse mechanisms, including membrane distortion[4], increased concentration of intracellular calcium or extracellular potassium[5,6], receptor-induced signalling pathways[7–9] and proteolytic cleavage of the distal C terminus[10–12]. Once activated, PANX1 channels generate voltage-dependent ionic current and allow permeation of large molecules such as fluorescent dyes (TO-PRO-3 and Lucifer Yellow) and nucleotides (ATP and UTP)[4,10,13]. PANX1-dependent release of nucleotides contributes to diverse (patho)physiological roles of PANX1, including cell clearance and inflammation[10,14], cancer progression[15,16], blood pressure regulation[8], metabolic defects[9] and neurological disorders[7,17,18].

In previous work, we described a PANX1 activation mechanism in which caspase cleavage of the cytoplasmic C terminus enables release of nucleotide 'find-me' signals, ATP and UTP, that attract phagocytes to apoptotic T lymphocytes for corpse clearance[10]. During apoptosis, the distal region of the PANX1 C terminus can be cleaved by caspases 3 or 7, releasing the pore-associated, autoinhibitory C-terminal tail (CT) to irreversibly activate the channel[10,11,19]. More recently, cleavage-based Panx1 activation at the same C-terminal site was observed during lipopolysaccharide-induced pyroptosis, in this case via caspase 11 (ref. 12). Despite clear demonstration of this cleavage/activation mechanism, it is not known how many C-tails must be removed to achieve PANX1 activation. Moreover, the associated changes in channel conformation and pore structure, and the corresponding effects on channel activity, remain to be elucidated.

In the current study, we used electron microscopy to show that caspase cleavage of the C-tail yields a capacious central pore. The fully activated conformation of PANX1 displays an outwardly rectifying unitary conductance ($<100$ pS maximum) that accounts for voltage dependence of PANX1 current. Furthermore, we find that progressive removal of C-terminal autoinhibitory regions leads to stepwise channel activation, with graded effects on unitary properties (single-channel conductance, open probability), dye uptake and ATP release. This stepwise, quantized progression is also observed with $\alpha1$ adrenoceptor-mediated PANX1 activation. Overall, our results demonstrate that sequential C-tail removal from individual subunits in hexameric PANX1 channels imparts distinct characteristics on the open conformation, controlling a common gate that coordinately regulates cell permeation of both small ions and large molecules to allow 'tunable' control of cell function and signalling.

## Results

**PANX1 pore revealed by caspase cleavage-mediated activation**. Caspase-mediated removal of PANX1 C-terminal autoinhibitory regions leads to channel opening, as measured by membrane currents and permeation of molecules such as ATP[10,11]. We used electron microscopy (EM) and single-channel recording of full-length and caspase-cleaved PANX1 to determine how C-terminal cleavage alters channel structure and function. After expression in Sf9 cells, purified full-length and caspase 3-cleaved PANX1 formed homogenous, thermostable oligomers, with elution volumes by size-exclusion chromatography (SEC) consistent with a predominant hexameric conformation (Supplementary Fig. 1a). Electron micrographs obtained from negatively stained samples were processed to obtain two-dimensional (2D) class-averaged images, with or without imposed six-fold symmetry[20] (Fig. 1a). In three independent determinations using different

image samples from full-length and caspase-cleaved PANX1 channel, class averages were obtained with a ring-shaped appearance that appeared to represent two different *en face* orientations (Fig. 1a). In one orientation, there was a small but obvious area of reduced density at the centre of the structure, presumably the pore (Fig. 1a); this orientation likely reflects a view from the extracellular face, since it was similar for both full-length and caspase-cleaved channels, even after cleavage at a cytoplasmic site. In the other orientation, however, there was a major difference between full-length and caspase-cleaved channels: a strikingly pronounced area of reduced central density was visible only in caspase-cleaved PANX1 (Fig. 1a). This more prominent 'pore' region was not seen in any of the class averages obtained from the full-length channel, despite systematically varying the numbers of classes used for averaging from 8 to 100 in three independent analyses. The larger 'pore' that appeared after caspase treatment suggests that this orientation represents a view from the cytoplasmic side of the channel, where the caspase site is located, and supports evidence that cleavage activates PANX1 by removing a pore-associated C-terminal autoinhibitory region[11]. Of note, the 'pore' structure observed after caspase cleavage is more pronounced than reported for channels exposed to high potassium[20], especially from the presumed cytoplasmic orientation.

We also used inside–out patch recordings to characterize single-channel properties of caspase-cleaved PANX1 in mammalian cells. We did not observe any PANX1-like channel activity immediately after excising inside–out membrane patches from HEK293T cells, consistent with our earlier findings that full-length human PANX1 channels are basally silent[10,11]. However, by exposing these previously silent patches to activated caspase 3, we could readily induce channel activity that was sensitive to carbenoxolone (CBX), a known inhibitor of PANX1 channels (Fig. 1b). We made several surprising observations regarding the steady-state properties of the fully 'open' channels. First, caspase-cleaved channels displayed an outwardly rectifying current–voltage (I–V) relationship, with a maximal unitary conductance of $96.2 \pm 2.0$ pS at positive membrane potentials and $12.2 \pm 0.2$ pS at negative potentials (Fig. 1c,d), and no evidence for openings to sub-conductance states. This differs from the linear single-channel I–V reported for extremely high-conductance channels ($\sim500$ pS, with multiple sub-conductance states) recorded from Xenopus oocytes in high extracellular $K^+$ (ref. 4). Second, although PANX1 is often considered voltage-dependent, the channel activity (open probability, $P_O$) was not different over a broad range of membrane potentials (Fig. 1e), indicating that gating of caspase-cleaved PANX1 is actually voltage independent. These channel properties were not affected by the C-terminal epitope tag present initially on this full-length PANX1 construct because identical results were obtained with a wild-type (untagged) version of PANX1 (Supplementary Fig. 1b), likely reflecting the fact that the tag is removed by caspase cleavage. Moreover, similar channel properties were observed in cell-attached recordings from HEK293T cells expressing C-terminally truncated PANX1, although in this patch configuration the channel displayed a slightly smaller unitary conductance at positive membrane potentials ($\sim75$ pS; Supplementary Fig. 1c–e). Thus, PANX1 channels activated by caspase cleavage display a particularly prominent 'pore', voltage-independent gating, and an outwardly rectifying unitary conductance that is $<100$ pS at depolarized potentials.

Although considered a hexameric channel, previous analyses of PANX1 stoichiometry were performed *in vitro*, after detergent solubilization of the protein[21]. To establish the stoichiometry of channels resident in the plasma membrane, we used total internal

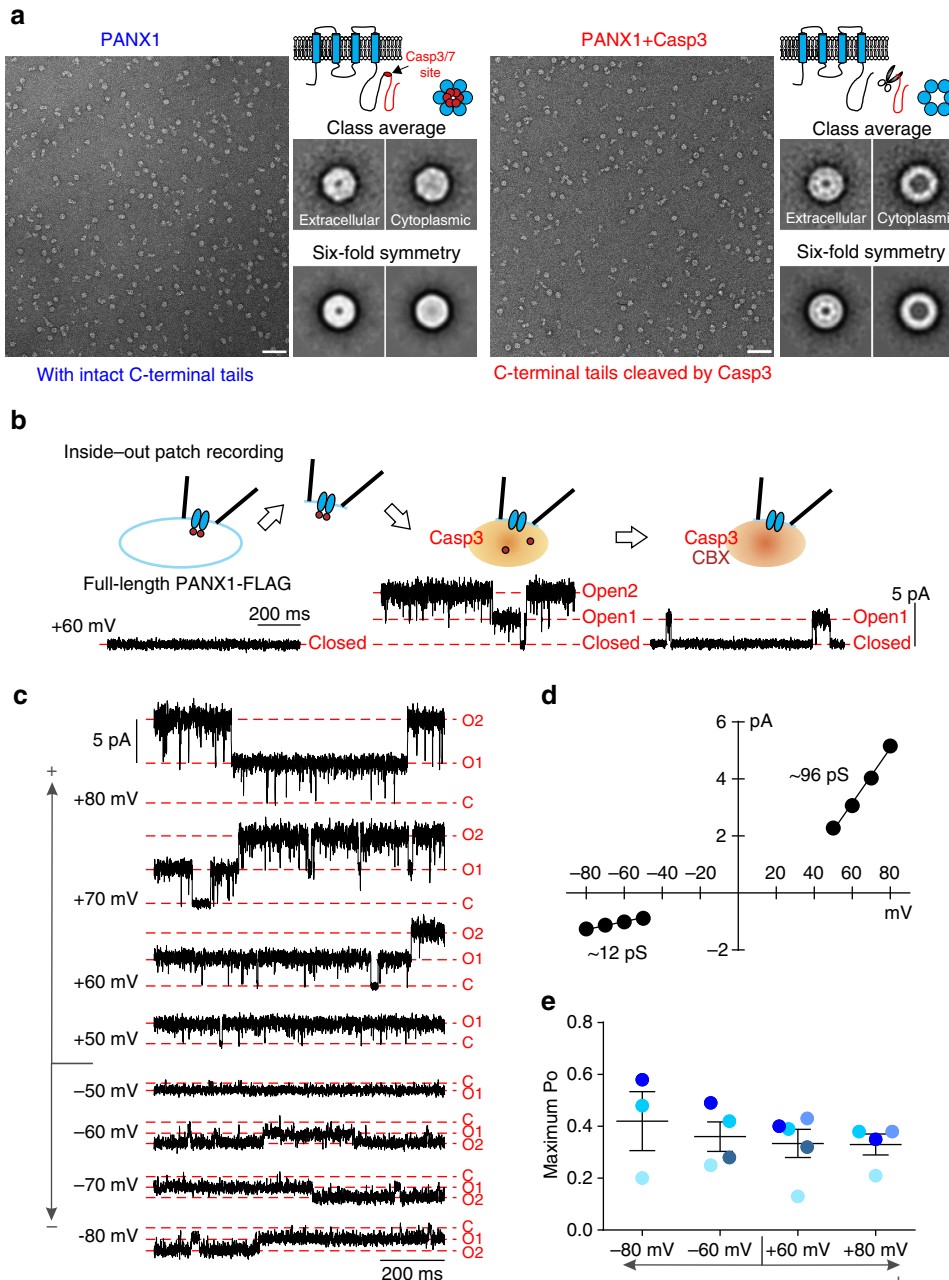

**Figure 1 | C-terminal cleavage by caspase 3 results in a distinctive PANX1 pore structure and maximum unitary conductance of ~96 pS.** (**a**) Electron micrograph and class-averaged EM images of negatively stained full-length (from 125 or 282 of 5,970 particles for the putative extracellular or cytoplasmic view, respectively) or caspase-cleaved PANX1 (from 79 or 56 of 6,892 particles); six-fold symmetry was imposed on the indicated images[20]. Schematics show the caspase cleavage site and expected cytoplasmic views of PANX1 hexameric channels, before and after cleavage. Scale bar, 50 nm; class-averaged image: 35.7 × 35.7 nm$^2$. (**b**) Inside–out recordings from HEK293T cells expressing full-length PANX1-FLAG following membrane excision, and after exposure to activated caspase 3 (Casp3) and carbenoxolone (CBX, 50 μM). (**c**) Steady-state activity of caspase-activated, full-length PANX1-FLAG in an inside–out patch held at different potentials. C, closed state; O1 and O2, open-state amplitude for one and two channels. (**d**) Averaged single-channel current amplitudes at different patch potentials (± s.e.m., smaller than symbol size) reveal an outward-rectifying unitary conductance: 96.2 ± 2.0 pS from +50 to +80 mV (n = 5) and 12.2 ± 0.2 pS from −80 to −50 mV (n = 4). (**e**) Open probability of cleavage-activated PANX1-FLAG is independent of membrane voltage. Data from each patch is represented by a different colour.

reflection fluorescence (TIRF) microscopy and single-molecule photobleaching[22]. The reliability of TIRF photobleaching is reduced with increasing numbers of photobleaching steps (≥5)[22], as would be expected for hexameric channels, so we prepared concatenated channels as a way to reduce the number of steps. We generated dimeric or trimeric PANX1 concatemers, in which individual subunits were covalently linked

in a head-to-tail order; the intervening linkers include a FLAG tag (DYKDDDDK) and Tobacco Etch Virus protease (TEVp) recognition sequence (ENLYFQG), and a single green fluorescent protein (GFP) was appended to the C terminus of each concatemer (Fig. 2a). Each individual subunit within the linked concatemer was either full-length or truncated at its own C-terminal caspase recognition site[10]. We refer to these

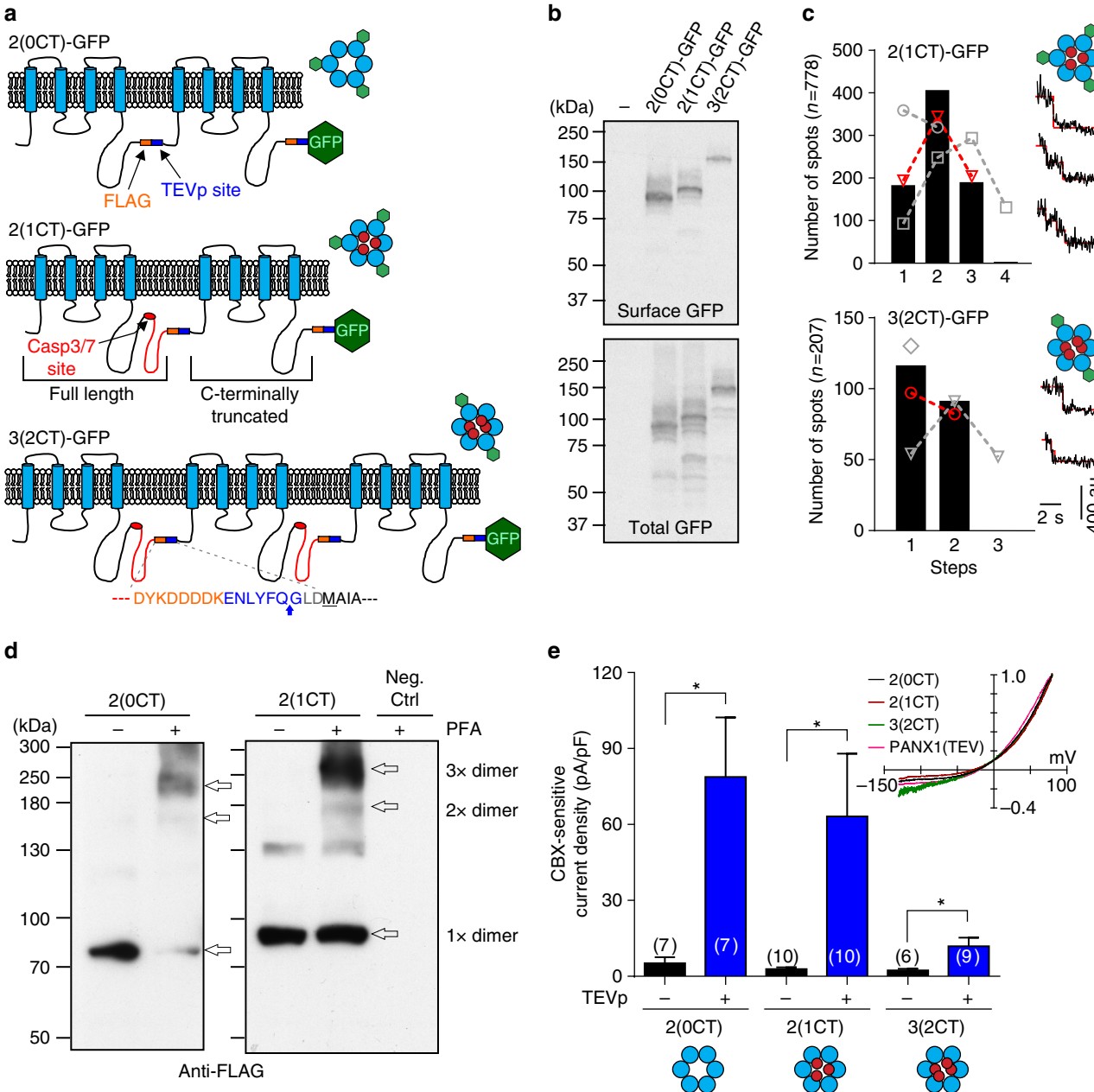

**Figure 2 | PANX1 forms hexameric channels in the plasma membrane.** (**a**) Schematic of GFP-tagged dimeric and trimeric PANX1 concatemers showing positions of FLAG tag (orange, DYKDDDDK) and TEV protease (TEVp) recognition site (blue, ENLYFQG) in linkers between PANX1 subunits; intact caspase site (red ellipse) is retained in full-length C-terminal domains (red). Linker sequence indicates TEVp cleavage site (blue arrow) and the start methionine (M). Expected hexameric conformations for the holo-channel are also provided. (**b**) Representative cell-surface biotinylation assay (from $n = 3$) shows GFP-tagged concatemers expressed on the plasma membrane (upper) and in the whole-cell lysate (lower). (**c**) Single-molecule photobleaching using TIRF microscopy. Histograms show number of spots with different photobleaching steps ($n = 3$ experiments; see insets for photobleaching examples). Note that only two particles from 2(1CT)-GFP showed more than three photobleaching steps and no particles from 3(2CT)-GFP showed more than two photobleaching steps. The lines and symbols overlaid on histograms represent fitted binomial distributions: tetramer (grey square), trimer (red triangle) or dimer (grey circle) for 2(1CT)-GFP; trimer (grey triangle), dimer (red circle) or monomer (grey diamond) for 3(2CT)-GFP. All fits assume ~65% of GFP fluorescence is available for photobleaching (Methods)[22]. (**d**) Cross-linked (paraformaldehyde (PFA)) dimeric PANX1 concatemers (no GFP tag) are primarily seen at a size expected for hexameric PANX1 (3 × dimer). (**e**) Whole-cell currents (mean ± s.e.m.) were observed from PANX1 concatemers in HEK293T cells only after inter-subunit linkers were cleaved by co-expressed TEVp. Inset: normalized I–V relationships for TEVp-activated, CBX-sensitive currents from monomeric (PANX1(TEV)) and concatemeric PANX1 channels are essentially identical. Number of recorded cells indicated. *$P < 0.05$ using two-way analysis of variance followed by Fisher's least significant difference (LSD) test.

constructs by the total number of incorporated subunits with the number of subunits containing a full-length CT in parentheses. Thus, a tandem dimer containing two covalently linked and C-terminally truncated PANX1 subunits is called

'2(0CT)'; a tandem dimer linking a full-length PANX1 subunit to a C-terminally truncated subunit is called '2(1CT)'; and a trimeric construct comprising two full-length PANX1 subunits together with a C-terminally truncated subunit is '3(2CT)'.

Following transfection in HEK293T cells, all GFP-tagged concatemers were expressed at the cell surface at the expected size (Fig. 2b). By analysing TIRF images from cells expressing 2(1CT)-GFP and 3(2CT)-GFP, we observed photobleaching steps that could be well fitted by binomial distributions consistent with either three 2(1CT)-GFP concatemers (that is, trimer of dimers) or two 3(2CT)-GFP concatemers (that is, dimer of trimers), supporting a hexameric PANX1 stoichiometry (Fig. 2c; Supplementary Fig. 2a). Paraformaldehyde-based protein cross-linking was consistent with the photobleaching experiments (Fig. 2d). To further demonstrate that these constructs were present on the cell membrane as hexamers, we combined protein cross-linking with surface biotinylation (Methods). After streptavidin precipitation, surface-expressed 3(2CT)-GFP constructs could be detected in a hexameric conformation ($2 \times$, a dimer of trimers; Supplementary Fig. 2b). Although it is difficult to quantify these cross-linking studies, the data are again consistent with PANX1 channels forming as hexamers on the cell surface.

We then determined the properties of currents from these dimeric and trimeric concatenated constructs. We were initially surprised that we did not find any CBX-sensitive current in cells transfected only with the concatenated PANX1 constructs (Fig. 2e, black bars). However, currents were readily observed when TEVp was co-expressed to cut the inter-subunit linker (Fig. 2e, blue bars). This suggested that the covalent linker between subunits interfered with channel function; it also provided a useful experimental system in which activity of concatenated channels can be strictly controlled via exposure to TEVp. In cells co-expressing TEVp, the 2(0CT) concatemer generated the largest PANX1 currents; in contrast, the 3(2CT) concatemer, with four full-length C-termini, produced the smallest PANX1 currents. Note that the $I–V$ relationships from concatemers were essentially identical to that from monomeric channels (Fig. 2e, inset). Also, the different current density between concatemers was not a result of differences in surface expression or glycosylation (Supplementary Fig. 2c,d). Collectively, results from dimeric and trimeric concatemers suggest that PANX1 channels can assemble as hexamers, and that PANX1 current increases when fewer C termini are present.

**C-terminal truncation yields stepwise PANX1 activation**. To define more precisely the quantitative and mechanistic relationships governing C-terminal effects on PANX1 channel activity, we performed inside–out patch recordings using a complete panel of hexameric PANX1 concatemers that contain 0–6 intact C termini; we call these 6(0CT) to 6(6CT), referring to the six concatenated subunits with the number of CTs within each hexameric unit provided in parenthesis (see schematic in Supplementary Fig. 3a). All hexameric concatemers were expressed in HEK293T cells, glycosylated and detected at the cell surface at their predicted molecular weights ($\sim 240–300$ kDa, Supplementary Fig. 3b–d).

Inside–out patch recordings were performed in HEK293T cells expressing hexameric PANX1 concatemers (Fig. 3a). On obtaining the inside–out configuration (Fig 3a, 1), the patch was exposed to purified TEVp to cleave the inter-subunit linkers and eliminate the linker-associated structural constraint in the concatenated channels (Fig 3a, 2). Single-channel currents were infrequently encountered prior to TEVp exposure in some patches. In a subset of those patches, we verified that this TEVp-independent channel activity was not due to PANX1 channels because it was insensitive to inhibition by CBX ($n=4$, data not shown); patches with those endogenous channels were not included. Cells were also treated with a pan-caspase inhibitor,

Q-VD-OPh ($20 \mu$M), during and after transfection, to prevent any endogenous caspase-mediated subunit cleavage that could disrupt the enforced stoichiometry. Thus, in the presence of TEVp, channel activity reflects that of the expressed hexameric concatemer with defined numbers of intact and truncated C termini.

As expected, we saw no channel activity from any of the hexameric PANX1 constructs before applying TEVp (Fig. 3b, left column). After TEVp exposure, channel activity became apparent for all constructs except 6(6CT), the concatemer in which each subunit retained an intact C terminus (Fig. 3b, centre column; Supplementary Fig. 4a). Remarkably, we noticed a progressive increase in single-channel current amplitude as the number of intact PANX1 C termini in the hexameric constructs decreased (Fig. 3b, centre column, top to bottom; Supplementary Fig. 4a); for each construct, channel openings were to a defined current amplitude, with no evidence for sub-conductance states. In addition, there was greater activity in constructs with fewer intact C termini, as those channels spent more time in the open state (see below for quantification of conductance and $P_O$). As exemplified by single-particle EM analysis of 6(0CT), the concatenated channel adopted an annular conformation with a prominent pore domain revealed by TEVp treatment, consistent with the functional data (Fig. 3c).

Following TEVp exposure, each patch was treated with activated caspase 3 to fully cleave any intact C termini remaining in the constructs; at this point, the concatemers are expected to be in an identical conformation, with all truncated and no intact C termini (Fig. 3a, 3). In accordance with this, the differences between the constructs were eliminated after caspase exposure (Fig. 3b, centre versus right column; also Supplementary Fig. 4a,b). The effect of this additional C-terminal cleavage is best seen with hexamers that initially contained the most intact C termini (for example, 6(4CT) to 6(6CT)). In the case of 6(6CT)-containing patches, which were silent after TEVp cleavage of the inter-subunit linkers, caspase treatment provided a crucial test to discriminate patches containing silent PANX1 concatemers from those simply devoid of channels.

We determined the unitary conductance for TEVp-cleaved PANX1 hexameric concatemers in each patch from the slope of single-channel current amplitudes between $+50$ and $+80$ mV (Fig. 4a); unitary conductance values for 6(6CT) through 6(0CT) were 0 pS ($n=3$), $24 \pm 3$ pS ($n=6$), $52 \pm 4$ pS ($n=9$), $70 \pm 4$ pS ($n=11$), $80 \pm 6$ pS ($n=4$), $91 \pm 4$ pS ($n=3$) and $96 \pm 8$ pS ($n=8$). After the remaining C termini were cleaved by caspase, all concatemers showed a similar conductance of $95 \pm 2$ pS ($n=28$, Fig. 4b). Evidently, there was no spurious effect of linking the subunits since the peak conductance obtained from fully cleaved, activated concatemeric PANX1 constructs was essentially identical to that of full-length, wild-type PANX1 subunits after caspase cleavage ($\sim 93–96$ pS; cf. Fig. 1d; Supplementary Fig. 1b). In addition, the same trend of increasing unitary conductance with reduced numbers of intact C termini was observed from dimeric and trimeric concatemers in inside–out patches after TEVp application, with similar conductance values (Supplementary Fig. 5). The relationship between unitary conductance and the number of cleaved C termini in the concatemers exhibits a Hill coefficient $<1$ ($h=-0.3$; Fig. 4c), suggesting that cleavage of each successive PANX1 C terminus increases channel conductance sub-linearly.

We also assessed steady-state open probabilities (NP$_O$) of the hexameric PANX1 concatemers to determine the effect of PANX1 C termini on channel gating. For comparison across constructs, and to account for patches that contained different numbers of active channels, we normalized the NP$_O$ obtained from each patch after TEVp application to

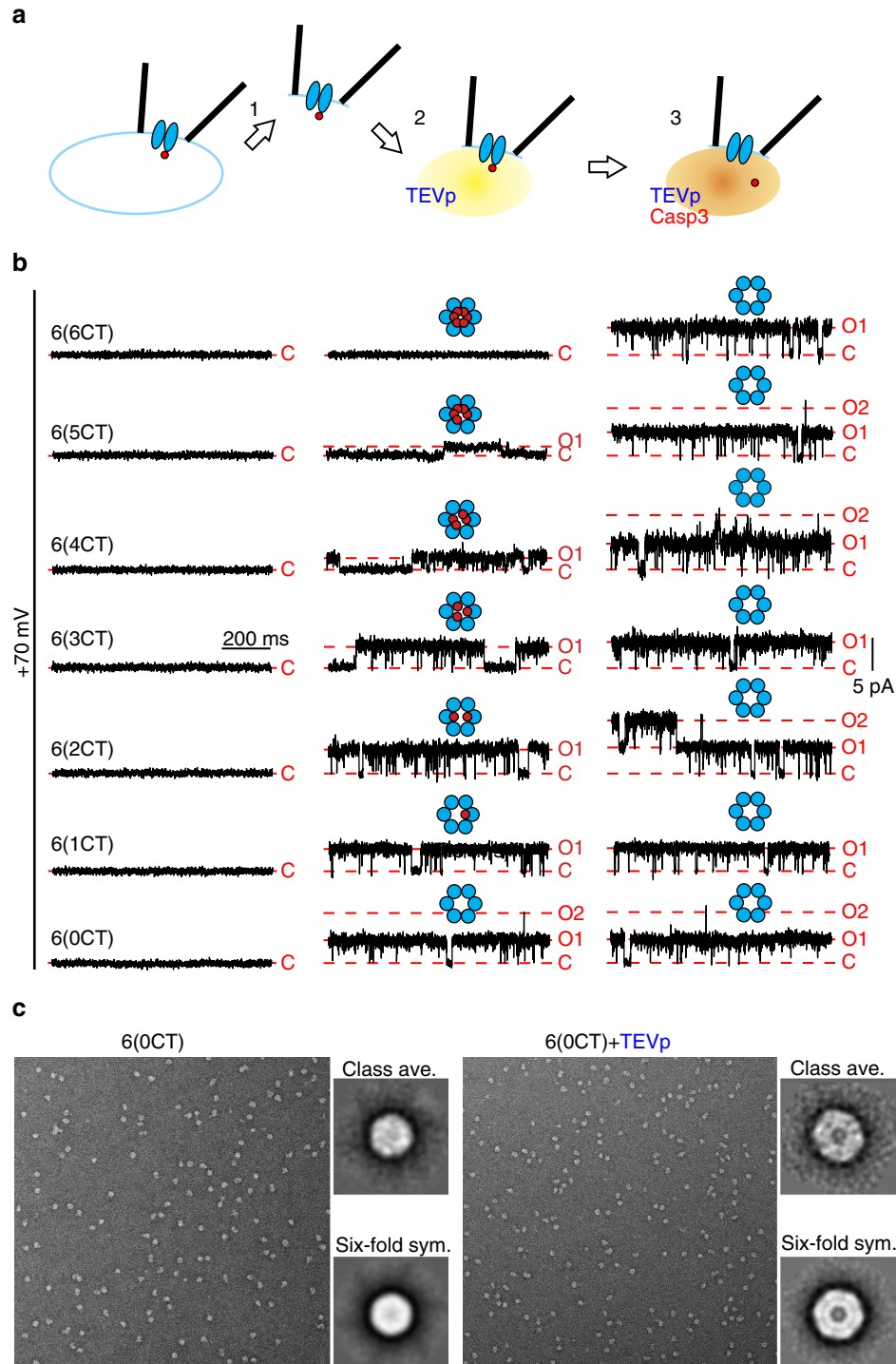

**Figure 3 | Inside–out patch recordings from HEK293T cells expressing hexameric PANX1 concatemers with different numbers of intact C termini.**
(**a**) Diagram illustrates protocol for inside–out patch recording. (**b**) Steady-state inside–out patch recordings obtained at +70 mV from HEK293T cells expressing hexameric concatemers with 0 to 6 intact C termini. (**c**) Electron micrograph and class-averaged images of negatively stained single particles of 6(0CT), before (left, 315 of 13,240 particles were used for class average) and after TEVp cleavage (right, from 38 of 5,883 particles). Images of imposed six-fold symmetry are provided. Scale bar, 50 nm; class-averaged image, $35.7 \times 35.7$ nm$^2$.

the NP$_O$ obtained from the same patch after caspase cleavage (Fig. 4d); this normalization recognizes that channel conformation is equivalent after caspase treatment, regardless of the C-terminal stoichiometry of the initial hexamer (that is, all constituent subunits have truncated C termini after caspase). We found that the open probability of PANX1 channel increased nearly linearly as more C termini were truncated (Fig. 4d).

Collectively, these results indicate that both conductance and gating of PANX1 channel are increased in a sequential manner by stepwise removal of the autoinhibitory CT. Although both activation processes occur progressively with C-terminal cleavage, the underlying mechanisms may be distinct. For conductance, cooperativity was observed, with initial C-terminal truncations having proportionally greater effects; for gating, each

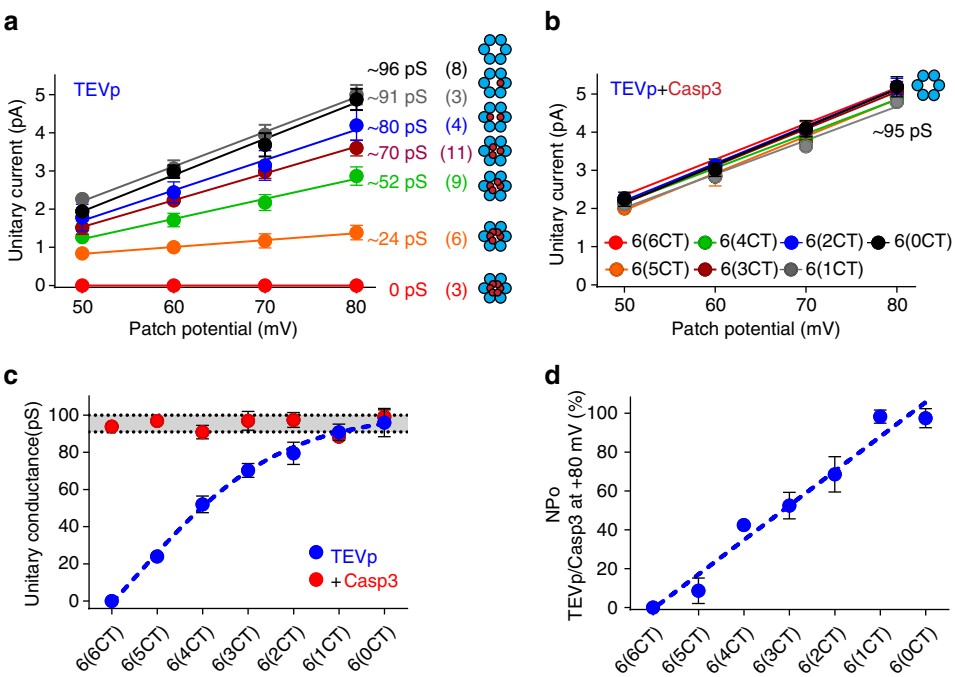

**Figure 4 | Decreasing numbers of PANX1 C termini increase unitary conductance and open probability of PANX1 channel.** (**a,b**) Single-channel *I–V* relationships from inside–out patches for PANX1 hexameric concatemers with different numbers of intact C termini recorded after TEVp exposure (**a**) and after removing remaining C termini with Casp3 (**b**). Unitary conductance indicated for each hexameric construct. Plotted data are mean ± s.e.m., and numbers of recorded patches are provided in parenthesis; for Casp3-cleaved concatemeric hexamers, conductance was ∼95 pS (*n* = 28). (**c**) Averaged unitary conductance (± s.e.m.) for each of the hexameric concatemers with different numbers of intact C termini (**a,b**), after TEVp exposure (blue) and from the same patches after Casp3 cleavage (red); data were fitted to a Hill equation, with coefficient -0.3 (blue dash line). The shaded area represents the 95% confidence interval for unitary conductance obtained from all fully cleaved concatemers (that is, after Casp3). (**d**) Open probability ($NP_O$; from + 80 mV) of PANX1 concatemers with different numbers of C termini (after TEVp) was normalized to that obtained from the same patches after Casp3 cleavage, averaged (± s.e.m.), and fitted by linear regression (blue dash line).

successive C-terminal cleavage event appeared to impart equivalent channel activation.

**Coordinate stepwise permeability to ions and large molecules.** We tested if the differences identified in single-channel properties of the hexameric PANX1 concatemers also manifest in distinct functional properties of those constructs at the macroscopic level, specifically examining whole-cell currents, ATP release and dye uptake. In whole-cell recordings, CBX-sensitive current was obtained from GFP-tagged PANX1 hexameric concatemers in HEK293T cells that were co-transfected with TEVp to cut the inter-subunit linkers (Fig. 5a). The 6(6CT)-GFP construct that includes all intact C termini generated no CBX-sensitive current, even when co-expressed with TEVp. For the remaining constructs, a stepwise increase in CBX-sensitive whole-cell current was observed as the numbers of C termini were progressively decreased (Fig. 5a). The different hexameric concatemers showed similar surface-to-total expression levels, as assessed by cell-surface biotinylation (Fig. 5b; see Supplementary Fig. 6 for uncropped images). In addition, normalized whole-cell *I–V* relationships were essentially identical for all PANX1 concatemers (Fig. 5a, inset), with reversal potentials in a similar range (between − 24 and − 28 mV), indicating that C-terminal truncation had negligible effects on rectification or ionic selectivity of PANX1 channels. Finally, for each of the hexameric concatemers, we compared experimentally acquired whole-cell currents (Fig. 5a, blue bars) to those predicted from single-channel properties (that is, the product of unitary conductance and open probability); this yielded a satisfyingly close correlation between microscopic and macroscopic channel behaviour (Fig. 5a, overlaid symbols).

We then used a luciferase assay to measure ATP release from HEK293T cells expressing the panel of hexameric PANX1 concatemers, with or without co-expressed TEVp. To prevent depletion of cytosolic ATP via TEVp-activated PANX1 hexamers, transfected cells were incubated with a reversible PANX1 inhibitor, trovafloxacin (Trovan)[23], and ATP concentration was measured 4 h after Trovan was removed in buffer containing ARL67156 (300 μM) to inhibit ecto-ATPase activity. PANX1-dependent ATP release was determined relative to parallel samples, in which the Trovan block was retained. Trovan-sensitive ATP release was not observed in cells that expressed either PANX1 concatemers or TEVp alone (Fig. 5c). In cells co-expressing TEVp with the concatemers, ATP release became apparent after 4 h with 6(3CT), then increased in a stepwise manner as the number of intact C termini was further decreased, reaching maximal levels in cells expressing 6(1CT) or 6(0CT) (Fig. 5c). As observed with whole-cell current, ATP release from hexameric PANX1 concatemers was well predicted by the cognate single-channel properties (Fig. 5c, overlaid symbols), which suggests a similar permeation pathway for small ions and ATP. We also extended this assay to include an 8-h collection period (Supplementary Fig. 7a). At this time point, we found clear evidence for PANX1-dependent ATP release from 6(4CT) that was elevated over that seen at 4 h time (5.4 ± 0.5 versus 0.6 ± 0.3 nM). However, we did not see any such release from 6(5CT)-expressing cells, even after 8 h. Thus, ATP flux through PANX1 channels requires removal of at least two C-tails, with progressively increasing permeation as additional C-tails are removed.

We used flow cytometry to examine TO-PRO-3 uptake in Jurkat T cells transiently transfected with the different

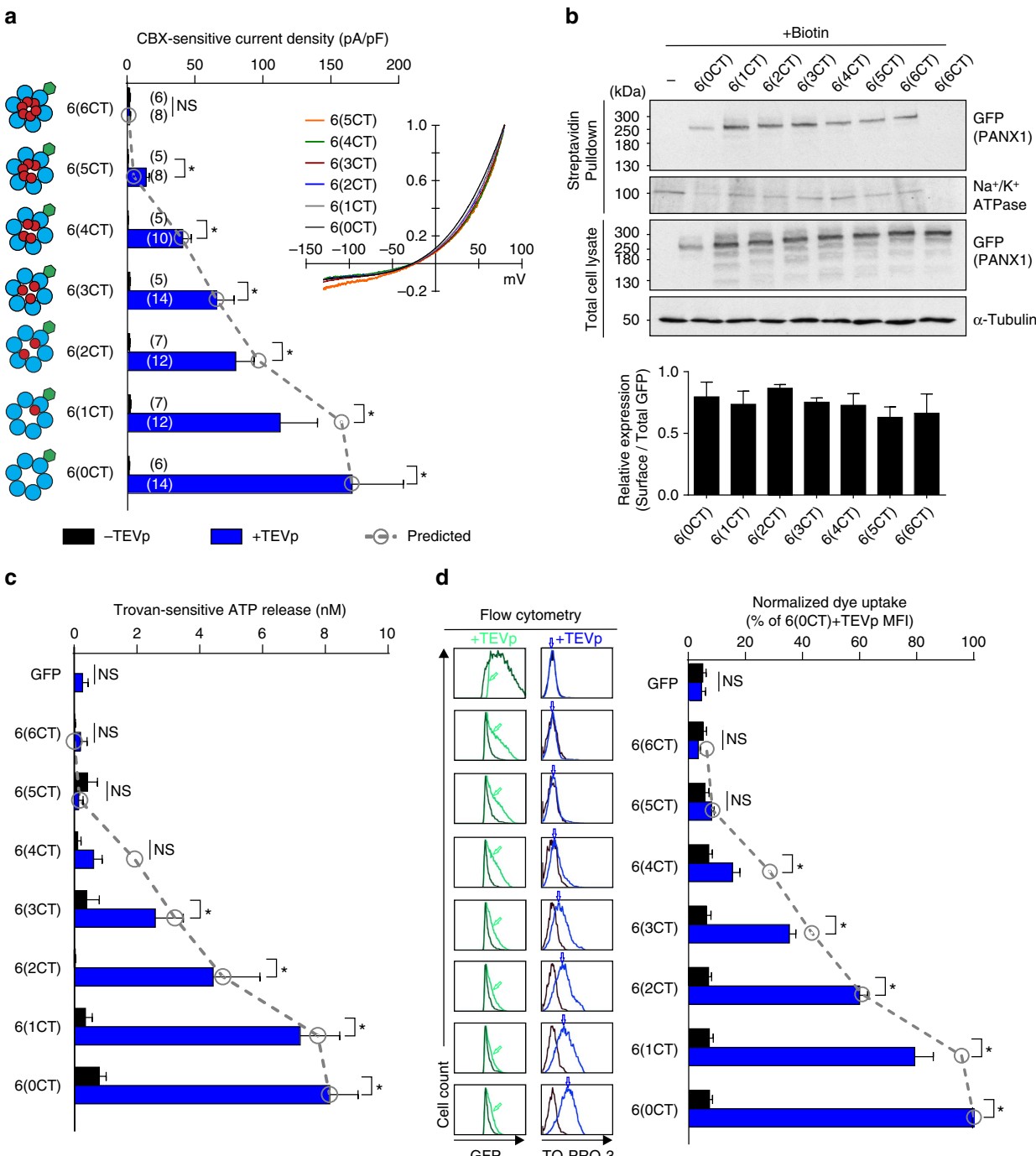

**Figure 5 | Graded increase in current and ATP/dye permeation via concatemeric PANX1 channels with decreasing numbers of intact C termini.**
(**a**) Whole-cell current density was obtained from HEK293T cells expressing various GFP-tagged hexameric PANX1 concatemers, with (blue) or without (black) TEVp co-expression. Numbers of recorded cells are provided in the parenthesis. Inset: averaged CBX-sensitive I–V relationships, normalized to current density at $+80$ mV, from cells co-expressing concatemers plus TEVp. Note that 6(6CT) is not depicted because that construct generated no CBX-sensitive current, even with TEVp co-expression. (**b**) Cell-surface biotinylation from HEK293T cells expressing GFP-tagged hexameric PANX1 concatemers. Representative Western blots (top) and grouped data (bottom, $n = 4$) show a similar ratio of cell-surface expression for all GFP-tagged PANX1 concatemers (normalized to total expression). Loading controls for streptavidin pull-down samples and total cell lysate samples were $Na^+/K^+$ ATPase and $\alpha$-tubulin. (See Supplementary Fig. 6 for uncropped images). (**c**) Averaged Trovan-sensitive ATP release from HEK293T cells expressing GFP-tagged hexameric PANX1 concatemers with (blue) or without (black) TEVp co-expression ($n = 4$ experiments; 4 h collection). (**d**) TO-PRO-3 uptake in Jurkat T cells expressing GFP-tagged hexameric PANX1 concatemers. Representative flow cytometry data shows GFP fluorescence (left) or TO-PRO-3 uptake (right) in cells with or without TEVp co-expression. Bar graph shows grouped data from four independent experiments; mean fluorescence intensity (MFI) of TO-PRO-3 uptake was normalized to the corresponding MFI for GFP within an experiment, and those values were normalized to the value for 6(0CT)+TEVp across independent experiments. Predicted whole-cell current (**a**), ATP release (**c**) and TO-PRO-3 uptake (**d**) based on single-channel properties are overlaid (grey data points). All data are presented as mean ± s.e.m.; *$P < 0.05$ using two-way analysis of variance followed by Fisher's LSD test; NS, no statistical significance.

GFP-tagged hexameric PANX1 concatemers (Fig. 5d). To avoid measuring dye uptake that could occur via activation of endogenously expressed PANX1 in Jurkat T cells[10], we limited our analysis to the cell population that was GFP positive (transfected), annexin V negative (non-apoptotic) and 7-aminoactinomycin D (7-AAD) negative (viable). This approach was successful in isolating dye uptake mediated by exogenously expressed concatenated PANX1 channels because fluorescence intensity above background was observed only in cells that were also co-transfected with TEVp (Fig. 5d). In Jurkat T cells co-expressing TEVp, the mean fluorescence intensity (MFI) of TO-PRO-3 was at background levels for the 6(6CT) and 6(5CT) constructs, and increased progressively as numbers of C termini in the channels were reduced from 6(4CT) through 6(0CT) (Fig. 5d, see representative flow cytometry data, blue peaks). To group data from multiple experiments ($n = 4$), the MFI of TO-PRO-3 was corrected for expression levels for each channel construct (that is, to the MFI for GFP) and normalized to the TO-PRO-3 MFI of 6(0CT) + TEVp for each experiment (Fig. 5d); this grouped analysis also showed that TO-PRO-3 uptake was undetectable in 6(6CT)- and 6(5CT)-expressing cells and increased progressively, as the number of C termini in the channel was reduced to four or lower (Fig. 5d, blue bars). Dye uptake was not observed in cells expressing 6(6CT) and 6(5CT) even when the assay period was extended to 60 min (Supplementary Fig. 7b). As observed with whole-cell current measurements and ATP release, when relative dye uptake was predicted based on measures of single-channel conductance and open probability, there was again a close correspondence between the experimental observations and the predicted values (Fig. 5d, overlaid symbols).

We note that there are some discrepancies between the predictions based on single-channel properties and the various whole-cell activities of the concatemers. The TEVp-activated 6(4CT) construct supported ionic current and yielded modest but measurable ATP release and dye uptake, albeit at apparently slower rates. By contrast, the 6(5CT) construct displayed small ionic currents, but it was not associated with any measurable ATP release or dye uptake even when assays were extended in time. This indicates that removal of a single C-tail is sufficient for ionic currents, but that elimination of at least two C-tails is required for ATP release and dye uptake, and supports the conclusion that different structural constraints govern permeability to small ions and large molecules.

**Progressive activation of caspase-activated native PANX1.** To test if native PANX1 channels exhibit a similar stepwise increase in unitary conductance during apoptosis, we carried out cell-attached patch recording in apoptotic Jurkat T cells and inhibited caspase activity at 30, 60 or 120 min of anti-Fas exposure by applying Q-VD-OPh (Q-VD, 25 μM). Indeed, we observed PANX1-like single-channel activities with different conductance levels in these patches. For example, in the cell-attached recordings depicted in Fig. 6a, two different CBX-sensitive channels were recorded with unitary conductance of 58 pS ($O_{S1}$) and 82 pS ($O_{L1}$); these were not sub-conductance levels of the same channel since transitions occurred from the closed state to each conductance level independently (to $O_{S1}$ or $O_{L1}$), and because simultaneous openings were observed (to $O_{S1} + O_{L1}$). In different patches from apoptotic cells, held between $+50$ and $+80$ mV, we found PANX1 channels with unitary conductance that ranged from ~20 to ~80 pS (Fig. 6b). The maximal conductance is similar to that observed from C-terminally truncated PANX1 expressed in HEK293T cells, also under cell-attached conditions (~75 pS; cf. Supplementary Fig. 1d). PANX1 channels with unitary conductance smaller

than the lower 95% confidence limit of that obtained from C-terminally truncated PANX1 (<66.2 pS, cf. Supplementary Fig. 1d) were observed more often in Jurkat T cells when caspase activity was inhibited by Q-VD after 30 min (13/21 channels) than after 60 min (6/13 channels) or 120 min (7/20 channels). These time-dependent effects on unitary conductance of caspase-activated PANX1 channels during apoptosis resemble the stepwise increase in unitary conductance in PANX1 concatemers with decreasing numbers of intact C termini (Figs 3 and 4).

**Graded conductance increase in α1DR-activated PANX1.** PANX1 channels can be activated by various Gαq protein-coupled receptors[24,25], including by α1 adrenoceptors that stimulate channel-mediated ATP release to regulate arteriolar resistance and blood pressure[8]; this α1 adrenoceptor mechanism is independent of caspase cleavage and can be recapitulated in HEK293T cells co-expressing α1D adrenoceptor (α1DR) and human PANX1 (Supplementary Fig. 8a). To test if PANX1 channels display graded changes in conductance when activated by this α1DR-mediated, cleavage-independent mechanism, we performed cell-attached patch recordings in this system before and after phenylephrine application (PE, 20 μM); we exposed patches to CBX in the presence of PE to reduce $P_O$ and more clearly visualize individual channel openings (Fig. 6c, schematic). As illustrated in an example record, channel activity became evident in previously silent cell-attached patches ~4 min after exposure to PE (Fig. 6c); at this time, we observed a channel with unitary conductance of ~23 pS ($O_{S1}$), the activity of which was reduced by subsequent exposure to CBX at ~8 min (Fig. 6c, boxed region). Another channel with larger conductance (~80 pS, $O_{L1}$) was observed at this time in the same patch (Fig. 6c, arrows). After prolonged exposure to PE and CBX, only the 80-pS channel activity remained in the patch (Fig. 6c, far right), and, as expected for PANX1, the open probability of this larger conductance channel increased dramatically after removing CBX from the bath (Supplementary Fig. 8b). PANX1 channels exhibiting various smaller conductance levels were also observed in other cell-attached patches shortly after PE exposure, with time-dependent transitions to larger conductance states before reaching maximal steady-state activation (Supplementary Fig. 8c,d). This suggests that α1 adrenoceptor-mediated PANX1 activation also occurs progressively, through open states with steadily increasing conductance.

α1 adrenoceptor-activated PANX1 channels display an outwardly rectifying unitary conductance at steady state, with properties similar to C-terminally truncated PANX1 (Fig. 6d). However, by comparison to cleavage-activated channels, α1DR-activated PANX1 have shorter mean open time (Fig. 6e–g). The flickering behaviour of α1DR-activated PANX1 is not due to the presence of sub-conductance states (Supplementary Fig. 8e, see data filtered at 5 kHz), but to short dwell time for the open state of channel (Fig. 6g). Despite differences in gating kinetics, these data indicate that activation of PANX1 channels by both C-terminal cleavage and α1 adrenoceptor signalling involves accumulative changes in open channel properties.

**Discussion**

In this study, we examined PANX1 activation by C-terminal cleavage and G-protein-coupled receptors and uncovered an unconventional, progressive activation mechanism for these multimeric channels. Specifically, stepwise increases in both unitary conductance and channel activity ($P_O$), accompanied by parallel increases in dye uptake and ATP release, followed from successive removal of distal C-terminal autoinhibitory regions from constituent subunits of the hexameric channel. Likewise,

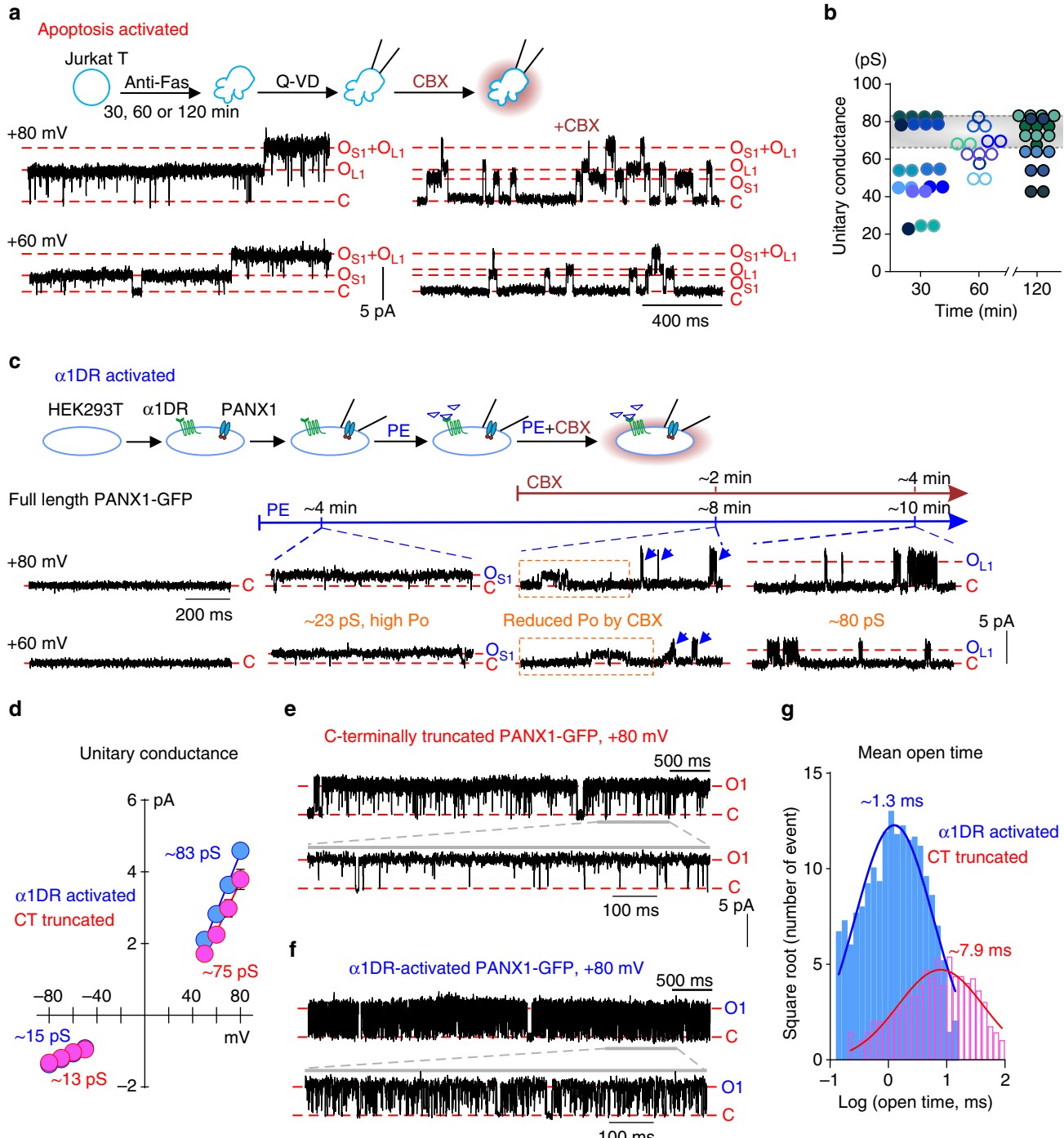

**Figure 6 | Single-channel activity of PANX1 is gradually increased during apoptosis and upon α1D adrenoceptor-mediated activation.** (**a**) Cell-attached recording from anti-Fas-treated apoptotic Jurkat T cells after inhibition of caspase activity at 60 min using 25 μM Q-VD-OPh (Q-VD); two CBX-sensitive channels with unitary conductance of either ∼82 pS ($O_{L1}$) or ∼58 pS ($O_{S1}$) were present in this patch. (**b**) Unitary conductance for CBX-sensitive PANX1 channels in apoptotic Jurkat T cells, obtained from cells treated with Q-VD at 30, 60 or 120 min of anti-Fas exposure. Data obtained from the same patch are labelled in the same colour. Shaded region represents 95% confidence interval of unitary conductance obtained from C-terminally truncated PANX1 in cell-attached configuration (from **d**; Supplementary Fig. 1d). (**c**) Cell-attached recording from HEK293T cells co-expressing α1D adrenoceptors (α1DR) and full-length PANX1-GFP channels obtained at +80 and +60 mV. PANX1 channel activity was not observed before phenylephrine (PE, 20 μM); small conductance channel openings ($O_{S1}$, ∼23 pS) observed after ∼4 min of PE exposure showed lower open probability after CBX (see boxed region). A larger conductance channel ($O_{L1}$, ∼80 pS) became predominant in the patch after ∼8 min exposure to PE (blue arrows). (**d**) Outwardly rectifying steady-state single-channel I–V relationships (mean ± s.e.m.) from cell-attached recordings of C-terminally truncated PANX1 (red) or α1DR-activated PANX1 (blue), with averaged unitary conductance of 74.7 ± 3.6 pS or 82.7 ± 1.4 pS at positive potentials (n = 8 and 4), and 12.8 ± 1.2 pS and 14.9 ± 2.1 pS at negative potentials (n = 4 and 3). (**e,f**) Representative cell-attached recordings at +80 mV from HEK293T cell expressing GFP-tagged, C-terminally truncated PANX1 (**e**) or α1DR and full-length PANX1 (**f**), ∼12 min after PE application. (**g**) Open time distribution from the cell-attached recordings depicted in **e,f**; mean open time for C-terminally truncated and α1DR-activated PANX1-GFP are ∼7.9 and ∼1.3 ms.

during activation by α1-adrenoceptors, PANX1 also transited through different channel conductance levels before ultimately attaining its maximal receptor-stimulated activity. These data indicate that discrete structural conformations associated with modulation of individual subunits impart distinct functional properties onto PANX1 channels, and activate a common path for permeating ions and large molecules. By tuning channel properties to differential subunit modification, it is possible to achieve variable intermediate levels of nucleotide release for ultrafine control of purinergic signalling (for example, with submaximal receptor occupation or subapoptotic caspase activation)[26].

On the basis of scanning mutagenesis, the distal C terminus of PANX1 was proposed to interact physically with the channel pore[27], and we previously demonstrated that dissociation from the pore is required for cleavage-mediated channel activation[11]. Thus, the striking decrease in optical density of the central pore of PANX1 observed after removal of the C-tails, either after caspase treatment of the full-length channel or TEVp treatment of the 6(0CT) hexameric concatemer, could reflect a physical displacement of C-tails from the channel pore. Indeed, unitary conductance and ATP/dye permeation increased with each successive C-terminal truncation, as if the presence of C-tails within the pore of the intact channel impeded transit of ions and large molecules. Consistent with such a steric effect, ionic current was observed with removal of only a single C-tail, whereas the larger ATP and dye molecules could permeate only after two C-tails were eliminated. Moreover, the *I-V* relationship and reversal potential (that is, ion selectivity) were similar among active PANX1 concatemers with different numbers of C-tails, suggesting that removal of PANX1 C termini likely changes resistance through the channel pore (for example, by increasing pore diameter or reducing the length of the permeation pathway) without affecting specific permeant-pore interactions. It is also worth noting that each particular C-terminal stoichiometry yielded its own stable unitary conductance, with no evidence for even transient sojourns to the conductance states associated with other C-terminal stoichiometries, implying that these hypothesized C-terminal pore associations must also be stable.

In heteroligomeric channels, unique conductance properties can be dictated by subunit composition to allow tuning of functional properties, as demonstrated recently for the related SWELL1 channel[28]. Our results show that single homomeric PANX1 channels can achieve functional tuning in a different manner, not by assembly from distinct subunit isoforms but rather by variable cleavage or modulation of constituent subunits. In most cases where channel activation is progressive and associated with sequential subunit modification (for example, $Ca^{2+}$ binding to BK channels, proteolysis-mediated activation of ENaC channels[29,30]), the increased activity involves changes in open-closed kinetics ($P_O$) without effects on unitary conductance. On the other hand, sub-conductance states of some ligand-activated multimeric channels may result from sequential conformational changes of individual subunits caused by successive ligand-binding events[31,32]. However, those specific sub-conductance states do not appear to be determined strictly by the number of occupied ligand-binding sites, since they can be visited when agonists are present at saturating concentrations or cross-linked to defined, but variable, numbers of subunits[31,33]. This contrasts with the specific activity levels and invariant conductance states for concatenated PANX1 channels with fixed C-terminal stoichiometry, a difference that likely reflects the nature of the caspase cleavage-based mechanism for channel activation that physically removes the C-terminal autoinhibitory region. Additional instances where channel properties are strictly determined by individual subunit modifications might also be revealed by using similar approaches to recapitulate sequential activation mechanisms.

Our data also suggest reappraisal of existing views on the properties of PANX1, a channel that is commonly considered to be voltage-dependent with extremely large conductance[4]. With respect to voltage dependence, whole-cell PANX1 currents from cleavage-activated channels show clear outward rectification that is mostly (if not entirely) accounted for by the rectifying single-channel conductance; since $P_O$ is unaffected by membrane potential, there does not appear to be any voltage-sensitive gating mechanism. With respect to PANX1 single-channel conductance, we found that two different forms of activation yield fully activated PANX1 channels that are < 100 pS, much less than that reported for the shorter isoform of PANX1 studied in Xenopus oocytes exposed to high extracellular $K^+$ ($\sim 500$ pS, with multiple sub-conductance states)[4]. In addition, ATP and dye permeation was observed with concatemeric channels that had even lower conductance levels. Moreover, due to single-channel rectification, the conductance is even smaller at negative membrane potentials experienced by cells studied for ATP release and dye uptake. Thus, there is not an obligatory requirement for PANX1 channels to attain an extremely high single-channel conductance state to support ATP release and dye uptake, or to exhibit a prominent 'pore' structure, as recently suggested[20]. The unitary properties were generally similar for PANX1 channels activated by either C-terminal cleavage or α1-adrenoceptor signalling, but distinct PANX1 channel properties could arise from different mechanisms of activation or alternative cell contexts.

In summary, our work reveals that each individual distal C terminus of PANX1 imparts a channel-intrinsic inhibition that limits unitary conductance and restrains channel gating. Although our studies focused primarily on proteolysis-mediated irreversible activation, similar reversal potentials, outwardly rectifying *I–V* relationships, and graded single-channel conductance were observed from PANX1 channels activated by a reversible α1 adrenoceptor-mediated mechanism. Thus, displacement of the C-tail may be a common process required for diverse mechanisms of PANX1 activation (Fig. 7). For cleavage-mediated PANX1 activation, we envisage that successive removal of the C-tail from each subunit progressively clears the permeation path and simultaneously supports enhanced channel activity. For adrenoceptor-mediated activation, sequential subunit modification may rearrange the positions of the C-tail to again allow increasingly larger conductance states and greater activity; the more flickering gating behaviour observed in adrenoceptor-activated PANX1 could ensue from effects of the retained C termini, either by flapping back to physically close the channel or by allosterically modulating the closed-to-open equilibrium. Although high-resolution structures are needed to ultimately reveal the precise disposition of the C-terminal autoinhibitory regions in PANX1 and to trace the sequential conformational changes that accompany channel activation, the data we present provide a new paradigm for quantized opening of multimeric channels. Given their homology to PANX1, we speculate that some of these features may be applicable to connexin, CALHM1 and SWELL1 channels, with significant implications for the cellular physiology and pathophysiology linked to this group of channels.

## Methods

**Reagents.** 7-AAD, TO-PRO-3 iodide (TO-PRO-3), EZ-link-sulfo-NHS-LC-biotin and disuccinimidyl suberate were obtained from Thermo Fisher Scientific. Annexin V-PE and Annexin V-Pacific Blue were obtained from Biolegend and Life Technologies, respectively. Electrode cuvettes (0.4 cm) were obtained from Bio-Rad, and 10× annexin V-binding buffer was obtained from eBioscience. All other chemicals were purchased from Sigma-Aldrich.

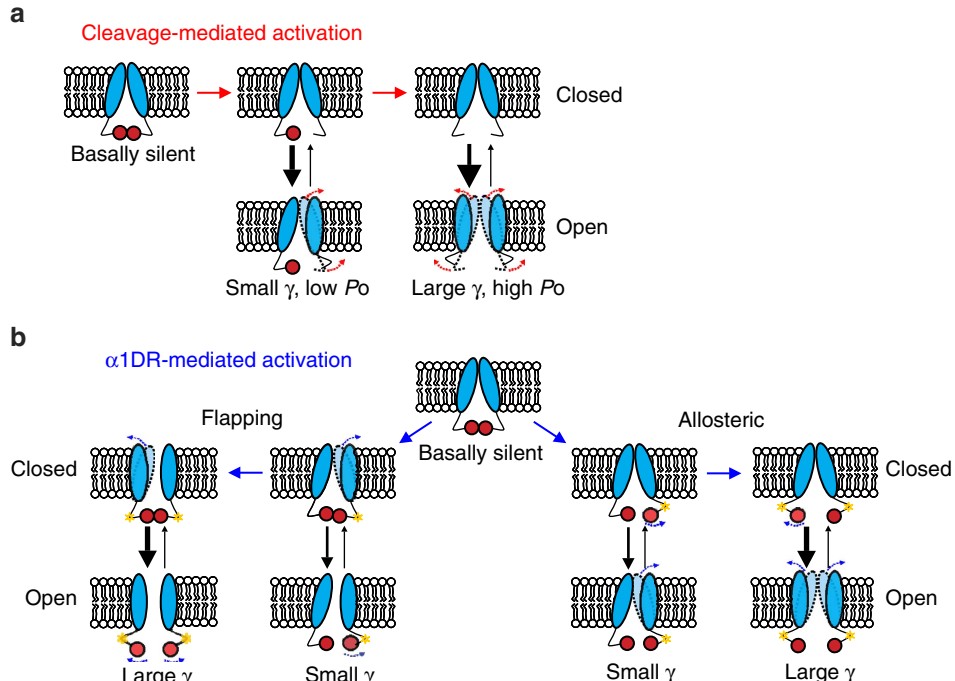

**Figure 7 | Hypothetical models for PANX1 channel activation. (a)** Schematics illustrate that individual cleavage at the C terminus of PANX1 subunits gradually increase unitary conductance ($\gamma$) and open probability ($Po$). (**b**) Post-translational modification (yellow star) of α1DR-activated PANX1 subunits gradually increases unitary conductance and open probability by either facilitating displacement of PANX1 C termini (left) or allosterically modulating channel gating via the displaced PANX1 C terminus (right).

**Expression and purification of PANX1 variants.** For PANX1 protein purification, full-length, human PANX1 (NM_015368, Isoform 1, 426 a.a.) modified with a enterokinase cleavage site (Asp-Asp-Asp-Asp-Lys) and a hexahistidine affinity tag at the carboxyl terminus (Supplementary Table 1, primers #1 and #2), was subcloned into pVL1932 vector (Invitrogen), followed by further subcloning into the pFastBac1 vector (Invitrogen; Supplementary Table 1, primers #3 and #4) for baculovirus expression in *Spodoptera frugiperda* (*Sf*9, Expression Systems) insect cells using the Bac-to-Bac expression system (Invitrogen). Recombinant PANX1 baculovirus was used to infect *Sf*9 cells at 27 °C with a density of $2 \times 10^6$ ml$^{-1}$ and MOI = 3. Cells were collected 48 h after infection by low-speed centrifugation at 2,000$g$. To isolate membrane-localized PANX1, *Sf*9 cell pellets were resuspended in low salt buffer A (50 mM HEPES, pH 7.5, 50 mM NaCl, 0.5 mM EDTA and protease inhibitor cocktails (Roche)) and lysed by Dounce homogenization ($\sim$30 strokes). Membranes were collected by ultracentrifugation at $100,000 \times g$ and washed with stepwise Dounce homogenization in low salt buffer A and high salt buffer B (50 mM HEPES, pH 7.5, 1 M NaCl, 0.5 mM EDTA and protease inhibitor cocktails) with ultracentrifugation at 100,000$g$ between steps. Final Dounce homogenization was performed in buffer C (50 mM HEPES, pH 7.5 and 500 mM NaCl), followed by ultracentrifugation at 100,000$g$. The membrane pellet was solubilized at 4 °C with 1% (w/v) *n*-tridecyl-β-D-maltopyranoside (TDM; Anatrace) in buffer D (50 mM HEPES, pH 7.5, 300 mM NaCl, 10 mM imidazole, 2.5% glycerol and protease inhibitor cocktails). Insoluble material was removed by ultracentrifugation at 100,000$g$, and the supernatant was incubated with $\sim$1.0 ml of cobalt-charged TALON metal affinity resins (Clontech) at 4 °C for 1 h. The resin was packed in an Econo-column (Bio-Rad, 1.0 × 10 cm) and washed with 10 mM imidazole, and 25 mM imidazole (20 column volumes/wash), and eluted with 250 mM imidazole in buffer C containing 0.02% TDM. Imidazole was removed using a G-25 buffer exchange column (GE Healthcare), and the eluted PANX1 proteins were concentrated to 1–2 mg ml$^{-1}$ using an Amicon ultracel-100 centrifugal filter unit (EMD Millipore). For generating C-terminally cleaved PANX1 proteins, purified full-length PANX1 was incubated overnight at 4 °C with recombinant, human caspase 3 (BD Biosciences) at a ratio of 1:500 (w/w). To improve contrast of negatively stained PANX1 particles, full-length and caspase-cleaved PANX1 were mixed with amphipol (A8-35, Anatrace) at a ratio of 1:3 (w/w) by gentle agitation for 4 h at 4 °C. TDM detergent was then removed by overnight incubation with Bio-Beads SM-2 (Bio-Rad) at 4 °C, and the Bio-Beads were subsequently removed using a disposable polyprep column (Bio-Rad). Aggregated material was removed by ultracentrifugation at 150,000$g$, and full-length or caspase-cleaved PANX1 were further purified by SEC on a SRT-C 300 column (Sepax Technologies). In addition to SEC, PANX1 purity was assessed by Simply Blue (SimpleBlue Safe Stain, Novex) staining of SDS–polyacrylamide gel electrophoresis (PAGE) using 4-20% pre-cast gels (Bio-Rad) and western immunoblot using anti-pentaHis antibodies (Qiagen). Thermal stability of PANX1

was characterized using a cysteine-reactive, coumarin-based fluorophore, CPM (*N*-[4-(7-diethylamino-4-methyl-3-coumarinyl) phenyl] maleimide[34].

**Cloning of PANX1 concatemers.** Wild-type human PANX1 (isoform 1), without an epitope tag, was made by inserting a stop codon proximal to the FLAG-tag of a previously described FLAG-tagged full-length PANX1 (ref. 10) (Supplementary Table 1, primers #5 and #6). To generate concatenated PANX1, FLAG-tagged and C-terminally truncated PANX1 were modified and subcloned individually into pcDNA3 (Invitrogen), as described below. A SalI site was first inserted at the 5′ end of full-length or C-terminally truncated PANX1 (in pEBB vector) using Quick-Change mutagenesis (Agilent; Supplementary Table 1, primers #7 and #8); a TEVp recognition site (ENLYFQG), XhoI site, and a stop codon were appended to the 3′ end of FLAG-tag sequence by PCR (Supplementary Table 1, primers #9–#12). Full-length or C-terminally truncated PANX1 in the pEBB vector were digested by SalI and XbaI, ligated at compatible XhoI and XbaI sites in the pcDNA3 vector to generate the first subunit of PANX1 concatemers. Note that SalI site at the 5′ end of PANX1 DNA fragment and XhoI site at the 3′ end of the pcDNA3 vector were eliminated by ligation. The XhoI site located 5′ to the stop codon in each successive PANX1 subunit allowed further concatenation by repetitive ligation as mentioned above. The sequence linking subunits in the concatemers is provided in Fig. 2a; note that the N terminus of the first subunit in all constructs is wild type, but a three amino acid addition (GLD) is present on the N terminus of subsequent subunits in the concatemers after TEVp cleavage.

To generate GFP-tagged PANX1, an enhanced GFP (eGFP) was C-terminally appended to either full-length or C-terminally truncated PANX1 using overlap-extension PCR. PCR products containing eGFP, with XhoI and XbaI sites at its 3′ end (Supplementary Table 1, primers #13 and #14), was amplified from pIRES2-EGFP (Clontech) using Pfu DNA polymerase (Agilent); PCR products containing PANX1 were separately amplified from the pEBB vector containing FLAG-tagged full-length or C-terminally truncated PANX1 using Pfu DNA polymerase (Supplementary Table 1, primers #15 and #16). eGFP with EcoRI and SalI sites to its 5′ end was appended to 3′ end of amplified full-length or C-terminally truncated PANX1 using overlap-extension PCR (Supplementary Table 1, primers #14 and #15). The overlap-extension PCR product was ligated to pcDNA3 vector by EcoRI and XbaI sites. These constructs were used for generating GFP-tagged PANX1 concatemers. T4 DNA ligase and restriction enzymes were obtained from the New England Biolabs. All constructs were verified by DNA sequencing.

**Expression and purification of PANX1 concatemers.** HEK293T cells (passages 8–15, American Type Culture Collection (ATCC)) were cultured at 37 °C

with humidified air containing 5% $CO_2$ in Dulbecco's Modified Eagle Medium (DMEM, high glucose, Gibco) containing 10% fetal bovine serum (FBS, Gibco), 1% penicillin and 1% streptomycin (PenStrep, Gibco). Transfections with GFP-tagged PANX1 concatemers lacking the CT (6(0CT)-GFP) were performed using polyethylenimine (PEI 25,000; Polysciences, Inc). After 48 h of transfection, cells were collected by centrifugation at 1500g. Membranes were isolated as described above and solubilized for 2 h at 4 °C with 1% (w/v) n-dodecyl-β-D-maltoside (DDM; Anatrace) and 0.2% cholesteryl hemisuccinate (CHS; Sigma) in 50 mM HEPES, pH 7.5, 300 mM NaCl, 2.5% glycerol and protease inhibitor cocktails. Unsolubilized material was removed by ultracentrifugation at 100,000g, and the supernatant was incubated overnight at 4 °C with Flag Affinity Resin (M2) (Sigma). After binding, the resin was packed in a Econo-column (Bio-Rad, $0.5 \times 5$ cm) and washed with 10 column volumes of 50 mM HEPES (pH 7.5), 300 mM NaCl, 0.2% (w/v) DDM, 0.04% (w/v) CHS, followed by 10 column volumes of 50 mM HEPES (pH 7.5), 1 M NaCl, 0.05% (w/v) DDM and 0.01% (w/v) CHS. The concatemer was eluted at 17 °C with 50 mM HEPES (pH 7.5), 500 mM NaCl, 0.02% (w/v) DDM, 0.004% (w/v) CHS containing the peptide DYKDDDDK (Bio Basic) at a concentration of 0.2 mg ml$^{-1}$. The purified 6(0CT)-GFP concatemer was deglycosylated with PNGase F (New England Biolab; 500 units per 1 h at 25 °C) before being mixed with amphipols (A8-35, Anatrace) at 1:3 (w/w) with gentle agitation overnight at 4 °C. The concatemer was then purified by fluorescence-detection size-exclusion chromatography (FSEC) on a Superose 6 10/300 column (GE Healthcare) attached to a fluorescence detector (Hitachi LaChrom Elite L-2485; $\lambda_{exc}$ of 488 nm and $\lambda_{em}$ of 509 nm for eGFP fluorescence; $\lambda_{exc}$ of 280 nm and $\lambda_{em}$ of 325 nm for tryptophan fluorescence) with a mobile phase of 50 mM HEPES (pH 7.5), 500 mM NaCl. Fractions containing the concatemer were collected and concentrated using a Vivaspin concentrator with a 100 kDa molecular weight cut off (GE Healthcare). The sample was further purified by repeat FSEC of the peak eGFP fluorescence fractions containing 6(0CT)-GFP concatemers using the Superose 6 10/300 column. For the 6(0CT)-GFP + TEVp sample, the concatemers were digested with TEV protease for 4 h at 17 °C before the final step of FSEC. Complete TEVp digestion was confirmed by SDS–PAGE and silver staining.

**Negative-stain and single-particle electron microscopy.** For negative staining, 3.5 μl of purified, monomeric PANX1 proteins (full-length or caspase-cleaved, 10–20 μg ml$^{-1}$) or 6(0CT)-GFP (before and after TEVp cleavage; 10–20 μg ml$^{-1}$) were applied to a glow-discharged, carbon-coated, 300-mesh, copper grid (Electron Microscopy Sciences) and stained with 2% uranyl acetate[35]. Low-dose EM was performed using a Tecnai F20 electron microscope (FEI), operating at 120 kV. Images were recorded at a nominal magnification of × 62,000 and a defocus of -0.9 μm using a $4 \times 4$ K charge-coupled device camera (UltraScan 4000, Gatan), corresponding to a pixel size of 1.82 Å on the specimen. EMAN2 software[36] was used for single-particle analysis. A total of ∼6,000 full-length and caspase-cleaved PANX1 particles from ∼150 micrographs were semi-automatically selected and extracted within boxes of $196 \times 196$ pixels using the Swarm tool in the e2boxer.py program of EMAN2. The contrast transfer function was estimated and corrected by e2ctf.py with standard EMAN2 parameters. Particle images were normalized and then high-pass (100 Å) and low-pass (10 Å) filtered and centred. The program e2refine2d.py was used to perform 2D classification by iterative, multivariate statistical analysis. Six-fold symmetry was applied to the 2D class averages using the program e2proc2d.py.

**Mammalian cell culture and transfection.** HEK293T cells (passage 14–30, ATCC) were cultured at 37 °C with humidified air containing 5% $CO_2$ in DMEM (high glucose, Gibco) containing 10% FBS (Gibco), penicillin, streptomycin, and sodium pyruvate and transfected using Lipofectamine2,000 (Invitrogen). HEK293T cells were transiently transfected with the selected GFP-tagged hexameric PANX1 concatemers, stained with Hoechst 33,342 (Molecular Probes), and imaged by using Axio Imager 2 (Zeiss) with a Plan-APOCHROMAT × 63 1.4 numerical aperture oil-immersion objective. Images were processed using Axio Vision Rel. 4.8 software (Zeiss). Jurkat T cells (E6.1, ATCC) were maintained in RPMI 1640 (Corning) with 10% FBS, penicillin, streptomycin, and L-glutamine at a density of 0.5–1.5 × 10$^6$ cells ml$^{-1}$. For transfection of Jurkat T cells, 5–10 × 10$^6$ cells were washed and resuspended in 400 μl plain RPMI 1640, followed by electroporation at 250 mV for 25 ms per pulse (BTX Electro Square Porator T820, Harvard Apparatus) with 5 μg of plasmids containing GFP-tagged PANX1 concatemers along with 10 μg of control vectors or TEVp-expressing plasmids. Transfected cells were studied 16–18 h after transfection.

**Subunit counting.** Subunit counting was performed by using single-molecule photobleaching on a Zeiss Axiovert 35 fluorescence microscope (Carl Zeiss) with a × 63 water immersion objective (Zeiss; numerical aperture = 0.95) and a prism-based TIRF illumination system as previously described[37]. HEK293T cells were plated on custom-made quartz slides 24 h before transfection and transiently transfected with GFP-tagged PANX1 concatemers ∼5–7 h before imaging. GFP-tagged PANX1 concatemers were illuminated by using a 488-nm laser (Innova 300C or OBIS 488 LX, Coherent) with a penetration depth of ∼100 nm. Fluorescence signals were recorded by an electron-multiplying

charge-coupled device camera (iXon DU-860E, Andor Technologies). The laser intensity, light-blocking shutters, and cameras were controlled by a homemade program written in LabVIEW (National Instruments). Images of $127 \times 127$ pixel$^2$ (corresponding to a sample area of $46.7 \times 46.7$ μm$^2$) were acquired with an exposure time of 50 ms, for a frame rate of 20 Hz.

Images were processed using ImageJ 1.48v (NIH, Bethesda, Maryland). First, the whole stack of images was subtracted by a heavily smoothed (rolling ball with radius of 64) copy of frames to correct variable background fluorescent intensity within the same frame and over a period time of photobleaching, and the maximum intensity for each pixel from a running average of the first 10 frames was projected on a single image. The centre pixel of a spot of $3 \times 3$ pixel$^2$ was identified from the running-average image, and mean fluorescence intensities from the selected spots were obtained from the background-subtracted images and plotted as a function of time. The number of photobleaching steps for each spot was determined manually. Photobleaching steps were defined by a drop in fluorescent intensity within less than two frames, between two dwell states where fluorescent intensity remained within noise levels for more than three frames[38]. If spots showed obvious movement or were too close to distinguish from other spots, they were excluded from further analysis. Only spots displaying discrete bleaching steps were included in analysis of binominal distribution. By using the Solver function of Excel and expected binomial distributions of trimer and dimer for 2(1CT)-GFP and 3(2CT)-GFP, respectively, the data were best fitted when the variable term for GFP available for photobleaching was ∼65% (ref. 22). Predicted binomial distributions for varying stoichiometry of PANX1 concatemers were then calculated by assuming that ∼65% of GFP was fluorescent (Fig. 2c) or by allowing the percentage of fluorescent GFP to vary (Supplementary Fig. 2a).

**Cell-surface biotinylation and protein cross-linking.** Cell-surface biotinylation studies were carried out in HEK293T cells transiently transfected with PANX1 concatemers. Transfected cells were labelled with 1 mg ml$^{-1}$ EZ-Link Sulfo-NHS-LC-Biotin (Thermo Scientific), in the presence of CBX (50 μM), in phosphate-buffered saline (PBS) for 1 h at 4 °C, and then incubated in cold PBS containing 100 mM glycine at 4 °C for 20 min to quench the reaction. Cells were lysed in PBS containing 1% Triton X-100 and a cocktail of protease inhibitors (Sigma-Aldrich), and cell lysates were incubated with streptavidin-agarose beads (Thermo Scientific) at 4 °C for 2 h to collect the biotinylated proteins. To examine if PANX1 concatemers were glycosylated, cell lysates were incubated with glycerol (50%) or PNGase F (7,000 U·mg$^{-1}$) at 37 °C for 3 h before streptavidin pull-down.

For protein cross-linking, transiently transfected HEK293T cells were fixed in PBS containing 0.4% paraformaldehyde for 12 min at room temperature and lysed as mentioned above. The solubilized proteins were collected by centrifugation at 4 °C, and the protein samples were mixed with 5 × Laemmli buffer (62.5% glycerol, 12.5% SDS, 0.5% bromophenol blue, 25% fresh 2-mercapto-enthanol in 30 mM Tris-HCl, pH 6.8), and immediately separated by SDS–PAGE after 5 min incubation at room temperature.

For cross-linking cell-surface proteins, HEK293T cells expressing PANX1 concatemers were first biotinylated at 4 °C for 30 min using 1 mg ml$^{-1}$ EZ-Link Sulfo-NHS-LC-Biotin in PBS containing CBX (50 μM). Biotin solution was removed and replaced with PBS containing disuccinimidyl suberate (1.8 mM, Thermo Scientific) at 4 °C for 1 h. The cross-linking was stopped in PBS containing 100 mM glycine for 20 min at 4 °C. Protein samples were extracted and prepared as above. Note that the data for cross-linking cell-surface proteins should not be considered quantitative, since the reactions engaged the same sites and thus the relative proportion of biotinylated and cross-linked protein could not be strictly controlled.

**Western blot analysis.** Protein samples were separated by SDS–PAGE, transferred onto 0.45 μm nitrocellulose membranes (PerkinElmer) and blocked with 5% non-fat dry milk dissolved in a tris-based buffer (10 mM Tris, 150 mM NaCl and 0.1% Tween 20, pH 7.4) at room temperature for 1 h. FLAG-tagged or GFP-tagged PANX1 concatemers were detected by incubating with anti-FLAG M2 (F3156, 1:1,000, Sigma-Aldrich) and anti-GFP (ab290, 1:7,000, abcam) antibodies. Anti-Na$^+$/K$^+$-ATPase (3010, 1:200, Cell Signaling) and anti-α-tubulin (T9026, 1:8,000, Sigma-Aldrich) antibodies were used as loading controls for cell-surface biotinylation and total cell lysate samples, respectively. Amersham ECL horseradish peroxidase-linked secondary antibodies (NA931V or NA9340V, 1:4,000 ∼ 1:8,000, GE Healthcare) and Western Lightning Plus-ECL (NEL103001EA, PerkinElmer) were used to visualize immunoreactive signals on Amersham Hyperfilm ECL (GE Healthcare).

**Purification of caspase 3 and TEV protease.** Recombinant caspase 3 precursors were prepared based on the previously described strategy[39]. In brief, a DNA sequence encoding a pro-caspase 3 Δ28/175TS deletion mutant was synthesized (GenScript) with 5′ NdeI and 3′ XhoI sites for insertion into the corresponding sites of pET-22b (+) (Novagen). Pro-caspase 3 plasmid-expressing Escherichia coli BL21(DE3) cells were treated with 1 mM IPTG at 18 °C for 18 h, and the pro-caspase 3 precursors were purified as described[39], with the exception that the cells were lysed using a microfluidizer. Purified pro-caspase 3 precursors were activated by thrombin[39]. TEVp was synthesized in E. coli BL21(DE3) CodonPlus-RIL cells

(Strategene) expressing the TEVp expression vector (pRK793), and purified as previously described[40].

**Electrophysiology.** All voltage-clamp recordings were carried out at room temperature; when recording from transiently transfected HEK293T cells, a pan-caspase inhibitor, Q-VD-OPh (Q-VD, 20 μM), was used during and after transfection to prevent inadvertent C-terminal cleavage of PANX1 by endogenously activated caspases. Micropipettes were pulled from thin-walled borosilicate glass capillaries (Harvard Apparatus) by using P-97 or P-87 puller (Sutter Instrument), and coated with Sylgard 184 silicone elastomer (Dow Corning Corporation).

Inside–out and cell-attached patch recordings were obtained using micropipettes with resistance of 7–10 MΩ and an Axopatch 200B amplifier controlled by pCLAMP10 software (Molecular Devices). Data were filtered to 5 kHz using an 8-pole low-pass Bessel filter (LPF-8, Warner Instruments) and digitized at a sampling rate of 20 kHz using a Digidata 1322A digitizer (Molecular Devices). Bath solution was composed of 140 mM NaCl, 3 mM KCl, 2 mM MgCl$_2$, 2 mM CaCl$_2$, 10 mM HEPES and 10 mM glucose (pH 7.3). Pipettes were filled with the bath solution as described, and $\geq 10$ GΩ seals were obtained in the bath solution. Patches were held at 0 mV before stepping to +50 to +80 mV and −80 to −50 mV (Δ10 mV, >5 s) for steady-state channel recordings; we did not observe a change in conductance or activity over time at test potentials. For inside–out patch recording, immediately after excising membrane patches, the bath solution was exchanged for an inside–out solution composed of 150 mM CsCl, 5 mM EGTA, 10 mM HEPES and 1 mM MgCl$_2$ (pH 7.3), and only those patches that were silent initially after excision were used (that is, those with no endogenous channel activity). Purified caspase 3 or TEVp was applied locally near the patch to a final concentration of ~1–2 μg ml$^{-1}$. For concatenated PANX1 channels, after steady-state activity was obtained (usually within 10 min after TEVp treatment), activated caspase 3 was applied locally (~1–2 μg ml$^{-1}$) and the steady-state channel activity was recorded. For cell-attached recordings, apoptosis of Jurkat T cells was induced by anti-Fas exposure for 30, 60 or 120 min, immediately followed by caspase inhibition using 25 μM Q-VD; HEK293T cells were transiently transfected with GFP-tagged C-terminally truncated PANX1, or with mouse α1DRs (MR222643, OriGene) and GFP-tagged full-length PANX1 channels. PE (20 μM) was applied in the bath solution with a constant flow to activate α1DRs. Single-channel activity was analysed using pCLAMP10; channel data were filtered to 2 kHz using a 8-pole low-pass Bessel filter for analyses of open probability and open time distribution, and filtered to 1 kHz for presentation except otherwise mentioned.

Whole-cell recordings were obtained at room temperature with 3–5 MΩ borosilicate glass patch pipettes and an Axopatch 200B amplifier in the bath solution described above. Internal solution contained 100 mM CsMeSO4, 30 mM TEACl, 4 mM NaCl, 1 mM MgCl$_2$, 0.5 mM CaCl$_2$, 10 mM HEPES, 10 mM EGTA, 3 mM ATP-Mg and 0.3 mM GTP-Tris (pH 7.3). Ramp voltage-clamp commands were applied at 7-s intervals using pCLAMP software and a Digidata 1322A digitizer. CBX-sensitive current was taken as the difference in current at +80 mV before and after CBX application, and was normalized to cell capacitance as a measure of current density.

**ATP release assay.** HEK293T cells were transiently transfected with hexameric PANX1 concatemers for 12 h, with or without TEVp contransfection. Cells were incubated with trovafloxacin (Trovan, 25 μM)[23] during transfection to prevent depletion of cytosolic ATP. Transfected cells were washed in an assay buffer (DMEM with 1% bovine serum albumin) containing either DMSO or Trovan, followed by incubation in the assay buffer containing ARL67156 (300 μM) and either DMSO or Trovan for 4 or 8 h. ATP in the supernatants of transfected HEK293T cells was measured using a luciferase/luciferin assay (CellTiter-Glo, Promega) according to manufacturer's instructions. Trovan-sensitive ATP release was obtained by subtracting the value for ATP concentration of the cells treated with Trovan from that of the cells exposed to DMSO. Cells were also incubated with Q-VD (25 μM) during and after transfection to prevent cleavage of intact PANX1 C termini. Increasing background ATP accumulation was observed with the more active hexameric constructs during the 8 h collection period, rendering this longer time point unreliable for quantitative comparisons across all constructs.

**Dye uptake assay.** TO-PRO-3 uptake analyses were performed using Jurkat T cells 16–18 h after transient transfection as mentioned above. Cells were stained with TO-PRO-3, 7-AAD and annexin V-PE/Pacific Blue for 15, 30 or 60 min at room temperature and kept on ice. TO-PRO-3 uptake and cell-surface staining were analysed on a BD FACSCanto flow cytometer (BD). Gating and data analysis were carried out using FlowJo software (v.8, FlowJo). 7-AAD$^-$/GFP$^+$ cells were used for TO-PRO-3 uptake analysis to exclude necrotic cells (7-AAD$^+$) and cells that did not express GFP-tagged PANX1 cocatemers (GFP$^-$). Annexin V$^+$ cells (apoptotic) were also excluded to remove any cells, in which endogenous caspases were activated. Small particles with low forward scatter (FSC) were gated out and MFI of TO-PRO-3 signal was analysed within the 7-AAD$^-$/GFP$^+$/Annexin V$^-$ population (viable cells expressing concatemers).

**Data analysis and statistics.** Data are presented as mean ± s.e.m. Statistical analysis was by two-way analysis of variance, with *post hoc* comparisons made using Fisher's least significant difference (LSD) test; differences between groups were considered significant if $P < 0.05$.

**Data availability.** The authors declare that data supporting the findings of this study are available within the paper and its Supplementary Information files and from the corresponding author on reasonable request.

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

## Acknowledgements

C.B.M was supported by T32 GM007055 from the NIH and X.J. received a predoctoral fellowship from the American Heart Association (15PRE25560084). This work was supported by R01 HL48908 (to M.Y.) and by R01 GM107848 and P01 HL120840 (both to D.A.B. and K.S.R.).

## Author contributions

Y.-H.C., M.Y., K.S.R. and D.A.B. developed the concept and designed experiments. Y.-H.C. performed electrophysiology, cell-surface biotinylation, TIRF photobleaching analysis and protein cross-linking. X.J., S.A.L. and B.C.B. carried out protein purification, negative-staining and single-particle EM analysis. C.B.M. performed fluorescent microscopic imaging, dye uptake and ATP release assays. V.K. and L.K.T. advised and technically supported TIRF photobleaching analysis. Y.-H.C., C.B.M. and S.S. generated concatenated PANX1 constructs. L.K.T., M.Y., K.S.R. and D.A.B. supervised the experiments. Y.-H.C., M.Y., K.S.R. and D.A.B. conceptualized and wrote the manuscript, and all authors edited the manuscript.

## Additional information

**Competing financial interests:** The authors declare no competing financial interests.

