## [Peer review file · Nature Communications]

PEER REVIEW FILE

Reviewers' comments:

Reviewer #1 (Remarks to the Author):

Chiu et al demonstrate that pannexin 1 channels are sequentially activated by caspase cleavage of their c-terminals. The approach taken is to generate concatameric channels and precisely and selectively control the number of c-termini available to 'block' the channels. The approach is highly innovative and rigorously implemented. I commend the authors for their creativity and attention to detail. Additionally, the manuscript is very well written and flows logically.

I am however, having difficulty reconciling the authors reported maximal single channel conductance in light of works from other groups. Dahl's group clearly showed a >500 pS open human pannexin channel using inside-out patches from xenopus (FEBS letters, 2004). MacVicar's group demonstrated that oxygen and glucose deprivation opened an ~500 pS channel in rat neurons (Science 2006). This contrasts greatly with the present manuscripts <100pS pannexin channels. Although the authors don't discuss the endogenous neuron pannexin, they do suggest that it arises from the 'non-physiological' levels of K in the xenopus recordings. There are other notable differences however, in Bao et al (FEBS, 2004) the PANX1 channels are activated by holding the cell between -20 and +20 mV to facilitate activation and channels (reportedly) rapidly inactivated at more positive potentials. Can these technical differences account for the authors small conductance channels?

What is the single channel conductance of the concatamers in symmetrical high (i.e. 150 mM) K solutions? Since K is a known activator of pannexin, does this promote the high conductance mode? Additionally, does holding the cell near 0mV prior to depolarizing to +80 mV affect unitary current levels and / or the shape of the IV curve? Does combining high K and the 0 mV holding current unmask a large conductance channel mode?

The concatameric approach relies on the hexameric structure of pannexin. The data presented in the paper (EM, photobleaching) are largely supportive of the hexameric structure and confirms the initial reports from Sosinsky's group (JBC 2010). A negative control would be an important addition. Since dimeric and trimeric concatamers form functional hexamers, I would predict that a quad or hepta concatamer would not be capable of forming a functional channel, even after caspase cleavage. Does TEVp cleavage of a quad concatamer allow for functional channels to form? If so, this would suggest assembly in the plasma membrane and not in the ER as has been suggested for connexins. If functional non-

hexameric channels can be formed, does this alter the interpretation of the approach? What is the conductance of these channels (if they form)?

The comparison of ionic currents, dye-uptake and ATP release is interesting. The data presented support ion fluxes with only a single truncated CT (i.e. Fig 5a; 6(5CT)), but ATP flux and dye-uptake are not detected until there are 2 and 3 CTs removed, respectively. The issue is superficially addressed on pg 13 of the manuscript and suggested to possibly represent a technical issue. I think there is the potential for important physiological insight into the channel's functions if ion flow is controlled by single C tail and large molecule flux by at least 3. Is there detectable ATP release or dye-uptake in the 6(5CT) channels if you wait longer for accumulation of ATP or dye? Alternatively, potential ectoATPases could be present to degrade released ATP in the 4 hour period, which could be blocked to improve signal. This would suggest that all forms of the channel are capable of fluxing each molecule, or confirm that the limiting step is the assay. Otherwise, if there is no appreciable ATP or dye-flux, what are the implications for the physiology of the channel.

Reviewer #2 (Remarks to the Author):

PANX1 is a channel that mediates nucleotide release for purinergic signaling. Thus PANX1 is involved in important physiological processes. Chiu et al. compared the behavior of wt and engineered concatemeric PANX1 hexamers using electrophysiological methods. The hexameric conformation was documented by negative stain EM, fluorescence bleaching of different GFP constructs and crosslinking results. The authors discovered a sequential gating activation that was reversible (α 1 adrenoceptor-mediated), and irreversible (caspase cleavage-mediated). Concatenated constructs contained 0-6 full-length c-termini and required TEVp to cleave the linkers for channel activation. These constructs allowed to characterize the effect of the c-termini. With increasing numbers, the channel unit conductance and the dye uptake rate was reduced in steps. The authors speculate that this quantized gating may be a generalized regulatory mechanism for other multimeric channels such as connexins, SWELL1 and CALHM1.

This is an interesting paper and it presents a new aspect of PANX1 regulation. The experimental approach is sound and mostly state of the art. Results are clearly presented and well documented.

Comments

EM data in Fig. 1 do not really show that PANX1 channels are hexamers. What is striking is the large cavity after caspase treatment. The concatenated dimeric and trimeric GFP-labelled constructs document the existence of PANX1 hexamers more convincingly. The number of particles within the class average should be indicated, as well as the total number of particles submitted to class averaging.

The statement 'this was confirmed in a subset of patches, for which TEV-independent channel activity was unaffected by CBX (n=4, data not shown)' is not clear. In Fig. 1 a distinct effect of CBX is seen only after caspase treatment.

Fig. 1a & 3c: Please indicate the number of particles in the class average (out of how many in total).

The TEV treated class average in Fig. 3c is about 20% larger than all other class averages. If significant it might be of interest; possibly only because of the selected particles.

Minor:

Error bars are missing in Fig. 6b & d.

Reviewer #3 (Remarks to the Author):

The study by Chiu and colleagues investigates that mechanisms linked to caspase-cleavage induced opening of PANX1 channels with a lesser focus on α_1 adrenoceptor mediated

activation of pannexin channels. Here the authors perform structural analysis to assess the open and closed states of PANX1 channels and electrophysiological measures to integrate wild-type and concatemeric channels. The authors confirm a previous conclusion from the Sosinsky laboratory (JBC 285:24420) that pannexin 1 channels are hexameric in nature. Most importantly the authors determine the PANX1 channels have discrete open states with single channel conductances of less than 100 pS and these states were linked to the involvement of a number of C-terminal domains in the hexamers.

This is a beautifully prepared and written manuscript that provides meaningful and insightful information on the gating and open states of PANX1 channels. I was surprised to see that the concatemeric approach worked as well as presented as this approach can lead to intracellular retention and misfolding of large protein complexes of this nature. While the hexameric nature of PANX1 is not new this study solidifies what has been generally accepted in the field for a number of years, albeit on a limited data set. Most importantly, this study challenges the position that PANX1 single channel conductances are of ~500 pS. Overall the study is detailed and exquisitely performed. However, there are some limitations in the study that need to be considered and some results need to be tempered. I am also not convinced that the authors have done enough on α_1 adrenoceptor mediated activation to warrant its inclusion in the title. This is primarily a dissection of the mechanisms underpinning caspase-based opening of PANX1 channels.

1) The authors need to more cautiously interpret the possible implications of C-terminal modifications of PANX1 as it carries a tri-glycine and hexahistidine affinity tag. This is particularly important given the implications of the C-terminal in regulating the channel. While I realize this is necessary for affinity purification it is also possible that this will adversely affect the channel conductance states through changes in the interactome or structurally important motifs. Similar caution must be exercised with the FLAG-tag and especially the 27kD GFP protein tag. No discussion is presented on how either tag may affect channel conductances. Are all these changes truly benign?

2) The use of PANX1 concatemers yielded some insightful results but the authors appear to have underappreciated how these complex proteins may undergo membrane insertion and oligomerization as super-sized channel proteins in tandem. While this approach has been used elsewhere for studying channels it has been associated with protein misfolding and premature protein degradation by quality control mechanisms. This is notable as the necessary Stop/Start sequences for membrane insertion through the translocon may be buried deep in the sequence. Furthermore, it is amazing that concatemers were functional when several copies of GFP were included in the channel. While it is clear they were, the authors need to consider how that may change the overall channel conductance states observed. It is hard to reconcile that it would not have any effect at all.

- 3) Tracings in Figure 1B shows only one incidence of the Open 2 state. What is the evidence that this is truly a PANX1 channel and how frequently is this open state seen for what I understand to be two channels? Likewise, Open 3 states are reported in the Supplementary data, yet no information is given as to how one can be sure these constitute 3 channels. Please explain.
- 4) The authors do not indicate what the glycosylation state is for any of the transfections and subsequent channel analysis. Based on the gel banding patterns seen in Figure 2 this is hard to determine but it appears that the concatemers may be glycosylated but is this assumption true? Since glycosylation has been reported to guide proper Panx1 trafficking to the cell surface, some assessment of glycosylation would be beneficial.
- 5) It is not clear to this reviewer what changes in the N-terminus (if any) of the PANX1 persists after TEVp cleavage? If the N-terminus is not truly native, could this also contribute to changes in channel conductances?
- 6) The Na/K ATPase blot in Figure 5B is not very convincing and this ATPase is totaling missing in the 6(6CT) lane for some reason?
- 7) It is not clear when the authors use wildtype PANX1 if it is truly unaltered and simply full length PANX1 with no tags or alterations. Please clarify throughout the manuscript. If it carries modifications, I would argue that it is no longer wild type. Also, there are now isoforms of PANX1 that have been reported. Which isoform is being used and is it possible that differences in reported channel conductances in the literature could be linked to the different PANX1 isoforms?
- 8) What does it mean that "All fits assume ~65% of GFP fluorescence is available for photobleaching"? How was this determined?
- 9) The authors must show some actual images of HEK293T cells that expressed the various PANX1 concatemers before and after TEVp co-expression? It is remarkable that these complex protein products are not mistargeted to some degree as the cell may recognize them as potentially misfolded protein complexes.
- 10) In the abstract and later in the manuscript the authors make reference to these findings possibly being applicable to connexins and other channel types. This should be removed as there is no evidence presented in the manuscript to support this notion or speculation. Specifically, the use of concatemeric connexins has been problematic and not particularly informative.

POINT-BY POINT RESPONSE TO REVIEWER COMMENTS

Reviewer #1:

General Comments

Chiu et al demonstrate that pannexin 1 channels are sequentially activated by caspase cleavage of their c-terminals. The approach taken is to generate concatameric channels and precisely and selectively control the number of c-termini available to 'block' the channels. The approach is highly innovative and rigorously implemented. I commend the authors for their creativity and attention to detail. Additionally, the manuscript is very well written and flows logically.

We are grateful for these comments on the quality and innovative nature of this work.

Main concerns:

1. *I am however, having difficulty reconciling the authors reported maximal single channel conductance in light of works from other groups. Dahl's group clearly showed a >500 pS open human pannexin channel using inside-out patches from xenopus (FEBS letters, 2004). MacVicar's group demonstrated that oxygen and glucose deprivation opened an ~500 pS channel in rat neurons (Science 2006). This contrasts greatly with the present manuscripts <100pS pannexin channels. Although the authors don't discuss the endogenous neuron pannexin, they do suggest that it arises from the 'non-physiological' levels of K in the xenopus recordings. There are other notable differences however, in Bao et al (FEBS, 2004) the PANX1 channels are activated by holding the cell between -20 and +20 mV to facilitate activation and channels (reportedly) rapidly inactivated at more positive potentials. Can these technical differences account for the authors small conductance channels?*

We understand the difficulty in reconciling our single channel conductance values from cleavage-activated channels in mammalian cells (<100 pS) with those published by Dahl's group on K-activated PANX1 channels recorded in Xenopus oocytes (>500 pS). In fact, we were also initially expecting to find these high conductance channels. However, that was not our experience with multiple PANX1 constructs in various cell systems where we find that the lower channel conductance is highly reproducible. As elaborated below, we do not believe that technical considerations account for these differences; in fact, we have so far been unable to observe K-mediated activation of PANX1 channels using two separate assay systems (PANX1 channel-mediated dye uptake and whole cell currents). For channels recorded in neurons, this is a bit tricky; the differences could reflect either a distinct mechanism of activation or misidentification of channels in native systems (especially possible given the imprecise pharmacopeia available for PANX1). For these reasons and to be respectful of the prior work, we stated our results cautiously in the original manuscript, even though we are completely confident in our data.

What is the single channel conductance of the concatamers in symmetrical high (i.e. 150 mM) K solutions? Since K is a known activator of pannexin, does this promote the high conductance mode?

We are not able to test this idea on single channel PANX1 properties because, in our hands, the channels are not activated by high extracellular K ([K]_e). First, our colleagues used a flow cytometry-based dye uptake assay and found that whereas UV-irradiated (i.e., apoptotic) spleno-

cytes readily concentrate TO-PRO-3 dye in a CBX-sensitive manner, those cells fail to take up TO-PRO-3 during exposure to the same high [K]_e levels used by Dahl's group (50 mM; unpublished observations of Michael Schappe & Bimal Desai, UVA Pannexin PPG Core 3). Second, [REDACTED] we directly tested the effects of high [K]_e on currents from wild type (un-tagged) or C-terminally deleted PANX1 channels in HEK293T cells; there was no effect of raised [K]_e on either of those channel constructs. Thus, there must be other factors in addition to elevated extracellular K that account for effects reported by Dahl and colleagues; we feel that sorting those out is beyond the scope of this work.

Additionally, does holding the cell near 0 mV prior to depolarizing to +80 mV affect unitary current levels and / or the shape of the IV curve? Does combining high K and the 0 mV holding current unmask a large conductance channel mode?

For our recordings, we typically held the patches at 0 mV before stepping to either positive or negative potentials to record channel activity. In addition, we did not observe a change in conductance (or activity) over time at depolarized potentials; thus, we have no evidence for inactivation of channels in our patches. This information has been added to the Methods (p. 27).

To summarize, we cannot replicate the high K-mediated PANX1 activation reported by Dahl and colleagues. A major conclusion from our work is that lower conductance states associated with physiological mechanisms of channel activation are compatible with both ATP release and dye uptake. Our data compellingly support this conclusion, and refute earlier suggestions that only the large (~500 pS) conductance state is capable of permeating ATP. Of note, our conclusion remains valid even if high [K]_e activates PANX1 channels to that larger conductance state. Although we point out in the paper that single channel properties after caspase and receptor activation are different from those reported for high K-activated channels (see p. 4, 17), we hope the Reviewer understands our preference to deal with other discrepancies regarding K-activation of PANX1 channels in a separate forum.

2. *The concatameric approach relies on the hexameric structure of pannexin. The data presented in the paper (EM, photobleaching) are largely supportive of the hexameric structure and confirms the initial reports from Sosinsky's group (JBC 2010). A negative control would be an important addition. Since dimeric and trimeric concatamers form functional hexamers, I would predict that a quad or hepta concatamer would not be capable of forming a functional channel, even after caspase cleavage. Does TEVp cleavage of a quad concatamer allow for functional channels to form? If so, this would suggest assembly in the plasma membrane and not in the ER as has been suggested for connexins. If functional non-hexameric channels can be formed, does this alter the interpretation of the approach? What is the conductance of these channels (if they form)?*

The Reviewer suggests an interesting control experiment with a (“quad”) tetrameric concatemer that would provide an additional test of the hexameric stoichiometry and/or evidence for plasma membrane assembly of subunits into functional channels.

We performed the suggested experiment (**Fig. R2**). For this, we chose to use an intracellular TEVp dialysis procedure, which avoided the concerns with distinguishing between non-functional channels vs. “empty” patches associated with inside-out recordings. We also chose to use a 4(oCT) construct in which all subunits have truncated C-termini; if these subunits rearranged on the membrane after TEVp-based linker cleavage to form a functional channel, then the resulting channel should be fully activated. Despite this, we observed no PANX1 currents under these conditions. In addition, we also co-expressed 4(oCT) with 2(2CT), testing if those constructs might combine either during biogenesis or while on the membrane into a TEVp-activated hexamer (i.e., with properties of 6(2CT)). Again, no currents were observed. We verified that the whole cell TEVp dialysis approach was effective using the TEVp-activated 2(oCT) construct (**Fig. R2c,d**; also **Fig. S5a**); we also verified

that the 4(oCT) construct was trafficked to the cell surface (**Fig. R2d, Inset**). These data do not support assembly of functional PANX1 channels from individual subunits on the cell surface.

The inability of a tetrameric construct to form functional channels is also consistent with a preferred hexameric conformation. However, at this point, we cannot explain why the “4(oCT) plus 2(2CT)” expression condition did not combine to yield functional channels (e.g., even though 2(oCT) or 3(2CT) can do so; see **Fig. 2e, Fig. S5**). Perhaps there is preferential assembly as trimers of dimeric constructs or dimers of trimeric constructs (e.g., due to favorable structural considerations) that are not shared by the tetramer. It would take substantial additional work to sort this out. In light of the uncertain interpretations of these results, the somewhat tangential nature of the question, and the potential for misinterpretation by a casual reader without access to or consideration of all the details, we hope the Reviewer appreciates our preference to leave these data out of this paper.

3. The comparison of ionic currents, dye-uptake and ATP release is interesting. The data presented support ion fluxes with only a single truncated CT (i.e. Fig 5a; 6(5CT)), but ATP flux and dye-uptake are not detected until there are 2 and 3 CTs removed, respectively. The issue is superficially addressed on pg 13 of the manuscript and suggested to possibly represent a technical issue. I think there is the potential for important physiological insight into the channel's functions if ion flow is controlled by single C tail and large molecule flux by at least 3. Is there detectable ATP release of dye-uptake in the 6(5CT) channels if you wait longer for accumulation of ATP or dye? Alternatively, potential ectoATPases could be present to degrade release ATP in the 4 hour period, which could be blocked to improve signal. This would suggest that all forms of the channel are capable of fluxing each molecule, or confirm that the limiting step is

the assay. Otherwise, if there is no appreciable ATP or dye-flux, what are the implications for the physiology of the channel.

We thank the Reviewer for these insightful comments, and have included new experiments that address the question of whether differences in dye uptake and ATP release among the hexameric constructs, especially 6(4CT) and 6(5CT), might be more apparent if the assays were extended in time (**Fig. R3**). We describe these results below, and provide the relevant new data and text in the revised manuscript. Specifically, we think this information allows a more definitive conclusion regarding structural constraints on permeation of large molecules via PANX1 channels.

First, note that ATP assays always include a blocker of ectoATPases (ARL67156, 300 μ M; see Methods, p. 28); we now also mention this in the text describing these data (p. 11).

We examined ATP release and dye uptake over a longer time scales (**Fig. R3**). For ATP release, an 8h collection period revealed clear PANX1-dependent (i.e., Trovan-sensitive) ATP release from 6(4CT) that is elevated over that seen at the 4h time point (5.4 ± 0.5 nM vs. 0.6 ± 0.3 nM, cf. Fig. 5c); we do not see PANX1-dependent ATP release from 6(5CT), even after 8 h. This suggests that removal of 2 C-tails can support ATP release, although at a reduced rate, whereas removal of a single C-terminus is not sufficient for ATP release. We did not use this later time point for quantification of ATP release in the main figure because it proved problematic for the most active constructs (e.g., 6(1CT) and 6(oCT)). That is, we saw greater cell death over this extended time frame with those highly active channels, leading to more ATP release that was not dependent on PANX1 (i.e., that was not Trovan-sensitive; see *red dashed line* in **Fig. R3**). Thus, the 4h time point we chose for quantification in Fig. 5c represents a compromise between maintaining cell viability while still observing PANX1-dependent ATP release from channels with lower activity (i.e., 6(4CT)).

For dye uptake, we extended the assay duration from 15 min. to 30 and 60 min., examining the 6(4CT) and 6(5CT) constructs in comparison to the fully activated 6(oCT); we included GFP alone or the inactive 6(6CT) construct as negative controls. TO-PRO-3 uptake above control was observed only with 6(4CT) and 6(oCT); a modest increase in mean fluorescence intensity (MFI) was seen at time points after 15 min. but relative levels of dye uptake (i.e., normalized to 6(oCT)) remained mostly stable. We did not observe any increase in dye uptake above control with 6(5CT), even after incubating with TO-PRO-3 for up to an hour. These data suggest that that dye uptake under these conditions is essentially complete after 30 min. and that removal of at least 2 C-tails is necessary for dye permeation.

In sum, results from these more protracted assays indicate that removal of 2 C-tails was necessary for both ATP release and dye uptake, i.e., ATP and dye permeation were observed with

Figure R3 (new Fig. S6). ATP release and dye uptake from select hexameric concatemers over extended assay time period. (a) ATP release was quantified by luciferase assay from HEK293T cells expressing the indicated PANX1 concatemers \pm TEVp after either release (DMSO) or maintenance of Trovan block of the channels for 8 h; PANX1-dependent ATP release is expressed as difference above background (i.e., DMSO-Trovan, see vertical arrows). Substantial PANX1-dependent ATP release was observed for 6(4CT) and 6(3CT) but not 6(5CT) at this time point. Note, however, that steadily increasing background ATP levels interfere with measures of PANX1-dependent release from the most active constructs (see red dashed line), likely due to cell death over this extended period of high channel activity. **(b)** We determined TO-PRO-3 uptake from Jurkat cells transfected with the indicated constructs for 15 to 60 min. Dye uptake was clearly seen with 6(4CT) and 6(oCT), the latter used as a positive control for normalization; although absolute values for mean fluorescence intensity (MFI) increased modestly after 15 min for both (*not shown*), the level of dye uptake for 6(4CT) relative to 6(oCT) was similar over all three assay time periods. Dye uptake was not observed above control with 6(5CT), or with either of the negative controls (GFP alone or 6(6CT)).

6(4CT) but not 6(5CT). This contrasts with ionic current, which can be observed with the 6(5CT) construct that is missing only a single C-tail. Note that differences in time course for the dye uptake (15-60 min) and ATP release experiments (>4h) likely reflect assay conditions (e.g., dye vs. ATP concentrations) and/or permeation rates of the molecules.

We have included these data in a new supplemental figure (**Suppl. Fig. 6**; see Methods, p. 28-29), and we describe the results (p. 11-12) and outline our interpretations of these results in more detail (p. 12, 16). In short, we suspect that structural determinants for flux of ions may be different than those for larger molecules, perhaps suggesting steric effects on permeation that reflect localization of the C tail in proximity to the pore. We again thank the Reviewer for this excellent suggestion.

Reviewer #2:

General Comments

PANX1 is a channel that mediates nucleotide release for purinergic signaling. Thus PANX1 is involved in important physiological processes. Chiu et al. compared the behavior of wt and engineered concatemeric PANX1 hexamers using electrophysiological methods. The hexameric conformation was documented by negative stain EM, fluorescence bleaching of different GFP constructs and crosslinking results. The authors discovered a sequential gating activation that was reversible ($\alpha 1$ adrenoreceptor-mediated), and irreversible (caspase cleavage-mediated). Concatenated constructs contained 0-6 full-length c-termini and required TEVp to cleave the linkers for channel activation. These constructs allowed to characterize the effect of the c-termini. With increasing numbers, the channel unit conductance and the dye uptake rate was reduced in steps. The authors speculate that this quantized gating may be a generalized regulatory mechanism for other multimeric channels such as connexins, SWELL1 and CALHM1.

This is an interesting paper and it present a new aspect of PANX1 regulation. The experimental approach is sound and mostly state of the art. Results are clearly presented and well documented.

We are grateful for this positive assessment of the work.

Comments:

1. *EM data in Fig. 1 do not really show that PANX1 channels are hexamers. What is striking is the large cavity after caspase treatment. The concatenated dimeric and trimeric GFP-labelled constructs document the existence of PANX1 hexamers more convincingly. The number of particles within the class average should be indicated, as well as the total number of particles submitted to class averaging.*

We agree that the large cavity after caspase treatment is particularly striking. We also agree that the combination of complementary approaches provides a more convincing demonstration of the likely hexameric nature of the channel than the EM data alone. We now indicate the number of particles for each class average and the total number of particles submitted to class averaging (see legends for Fig. 1a, Fig. 3c).

2. *The statement ‘this was confirmed in a subset of patches, for which TEV-independent channel activity was unaffected by CBX (n=4, data not shown)’ is not clear. In Fig. 1 a distinct effect of CBX is seen only after caspase treatment.*

We apologize for the confusion. In inside-out patches from cells transfected with concatenated channels, we did not see any PANX1 channel activity before application of TEVp. In a few cases, channel activity was evident before TEVp. However, we do not think those were PANX1 channels because, when tested, TEV-independent channel activity was unaffected by a PANX1 blocker (CBX, tested in 4 patches). We have now re-worded to clarify this statement (p. 7).

3. *Fig. 1a & 3c: Please indicate the number of particles in the class average (out of how many in total).*

See response to 1, above. We provide the number of particles for each class average and the total number of particles submitted to class averaging (in the legends for Fig. 1a, Fig. 3c).

4. *The TEV treated class average in Fig. 3c is about 20% larger than all other class averages. If significant it might be of interest; possibly only because of the selected particles.*

We are reluctant to emphasize this difference in the particle size, which may simply reflect differences in methods used for expressing and purifying the proteins. The particle size depends on the extent of “puddling” of the negative-stain around the particles, and the extent of stain penetration also depends on the detergent micelle surrounding the channels. Channels derived from monomeric subunits were expressed in Sf9 insect cells and those from hexameric concatemers were expressed in HEK293T cells. Also, the Sf9-generated proteins were solubilized in TDM and exchanged with amphipol before purification by SEC; those from HEK cells were solubilized in DDM/CHS before deglycosylation, exchange with amphipol and purification by FSEC. It is hard to predict how these methodological differences might affect the appearance of particle size in negatively-stained EM samples. Lastly, we note that one of the subunits has a GFP tag, which could affect the apparent diameter. For our purposes, we simply intended to show that the concatemer particles were not grossly misshapen (i.e., they retained an annular appearance), and that TEVp treatment of the 6(OCT) concatemer produced a large pore similar to that seen with caspase treatment of the monomer-derived channel protein.

Minor:

1. *Error bars are missing in Fig. 6b & d.*

In Fig. 6b, we intended to show the conductance of each individual channel in the patch, with data from each patch provided in a different color; there are no associated error bars since these are individual data points rather than averages. For Fig. 6d, the error bars are provided but they are often smaller than the symbol. In order to make the error bars more visible in Fig. 6d, we changed their color to black (instead of the symbol color, as originally presented.)

Reviewer #3:

General Comments

... The study by Chiu and colleagues investigates that mechanisms linked to caspase-cleavage induced opening of PANX1 channels with a lesser focus on $\alpha 1$ adrenoceptor mediated activation of pannexin channels. Here the authors perform structural analysis to assess the open and closed states of PANX1 channels and electrophysiological measures to integrate wild-type and concatemeric channels. The authors confirm a previous conclusion from the Sosinsky laboratory (JBC 285:24420) that pannexin 1 channels are hexameric in nature. Most importantly the authors determine the PANX1 channels have discrete open states with single channel conductances of less than 100 pS and these states were linked to the involvement of a number of C-terminal domains in the hexamers.

This is a beautifully prepared and written manuscript that provides meaningful and insightful information on the gating and open states of PANX1 channels. I was surprised to see that the concatemeric approach worked as well as presented as this approach can lead to intracellular retention and misfolding of large protein complexes of this nature. While the hexameric nature of PANX1 is not new this study solidifies what has been generally accepted in the field for a number of years, albeit on a limited data set. Most importantly, this study challenges the position that PANX1 single channel conductances are of ~ 500 pS. Overall the study is detailed and

exquisitely performed. However, there are some limitations in the study that need to be considered and some results need to be tempered. I am also not convinced that the authors have done enough on $\alpha 1$ adrenoceptor mediated activation to warrant its inclusion in the title. This is primarily a dissection of the mechanisms underpinning caspase-based opening of PANX1 channels.

We are grateful for these comments on the quality and importance of this work. We have now simplified the title to: “A quantized mechanism for pannexin channel activation.”

The Reviewer raises an overarching concern with potential limitations of the work, particularly regarding use of concatemers and tagged constructs, that s/he feels require some tempering of the conclusions. We address each of the specific comments below. However, we wish to point out in advance that we provide a number of controls and independent methods that validate our findings (including some new experiments performed in response to the Reviewer comments). We think these data collectively attest to the veracity of the results and the conclusions they prompt. We appreciate that the Reviewer pointed out these perceived limitations; by addressing these points in the revised manuscript, we believe the paper has been further strengthened.

Comments:

1. *The authors need to more cautiously interpret the possible implications of C-terminal modifications of PANX1 as it carries a tri-glycine and hexahistidine affinity tag. This is particularly important given the implications of the C-terminal in regulating the channel. While I realize this is necessary for affinity purification it is also possible that this will adversely affect the channel conductance states through changes in the interactome or structurally important motifs. Similar caution must be exercised with the FLAG-tag and especially the 27kD GFP protein tag. No discussion is presented on how either tag may affect channel conductances. Are all these changes truly benign?*

We thank the Reviewer for recommending caution. As recognized, the tri-glycine and hexahistidine affinity tag were necessary to purify the channel for the EM studies. Although it is possible that these tags could somehow affect the channel interactome, this is not expected to influence structural properties of the channel assessed by EM after purification. Also, as discussed below, we find no effect of C-terminal modifications on channel properties (which are actually removed during cleavage-based activation; please see response to Point #7). Specifically, we have now performed inside-out patch recordings with wild type PANX1 without any epitope tags, and find that the conductance properties are identical to the (initially) tagged versions of the channels (see **Suppl. Fig. 1b, p. 5**). In addition, cell-attached recordings from apoptotic Jurkat cells after more prolonged anti-Fas stimulation (120 min.) are now provided (see **Fig. 6b**); these new recordings of native PANX1 channels more fully cleaved by endogenous caspases show conductance properties that are consistent with recordings from the corresponding truncated, and tagged recombinant channels.

2. *The use of PANX1 concatemers yielded some insightful results but the authors appear to have underappreciated how these complex proteins may undergo membrane insertion and oligomerization as super-sized channel proteins in tandem. While this approach has been used elsewhere for studying channels it has been associated with protein misfolding and premature protein degradation by quality control mechanisms. This is notable as the necessary Stop/Start sequences for membrane insertion through the translocon may be buried deep in the sequence. Furthermore, it is amazing that concatemers were functional when several copies of GFP were included in the channel. While it is clear they were, the authors need to consider how that may change the overall channel conductance states observed. It is hard to reconcile that it would not have any effect at all.*

We are thankful that our efforts were not undone by these (or other) theoretical reasons that might easily have led to lack of expression or function of the concatenated constructs.

As for the suggestion that linking the subunits must cause some change in conductance, we offer our data as evidence to the contrary. Specifically, we find that caspase cleavage of channels formed from monomeric subunits yielded conductance values that were virtually identical to those obtained from any of the fully caspase-cleaved concatemers (both ~96 pS, cf. **Figs. 1 & 4**).

Note that there is only a single GFP at the C-terminal end of the final subunit in the concatemers (not several copies). Also note that the N-terminal end of the first subunit in the concatemers remains unmodified, which may explain maintained membrane insertion of concatemers.

3. *Tracings in Figure 1B shows only one incidence of the Open 2 state. What is the evidence that this is truly a PANX1 channel and how frequently is this open state seen for what I understand to be two channels? Likewise, Open 3 states are reported in the Supplementary data, yet no information is given as to how one can be sure these constitute 3 channels. Please explain.*

Fig. 1 now provides new traces that show more (and longer) openings to the indicated states.

It is a challenge to determine the overall number of channels in a membrane patch. However, we believe that each of these current amplitude levels represents opening of additional PANX1 channels because of the equal increment in unitary current amplitudes between states, including transitions from O1→O2 or from O3→O2. This is consistent with simultaneous openings of multiple channels, each with the same conductance properties; subconductance states of a single channel typically do not display such quantal increments (including those reported by Dahl's group for K-activated large-conductance PANX1 channels).

4. *The authors do not indicate what the glycosylation state is for any of the transfections and subsequent channel analysis. Based on the gel banding patterns seen in Figure 2 this is hard to determine but it appears that the concatemers may be glycosylated but is this assumption true? Since glycosylation has been reported to guide proper Panx1 trafficking to the cell surface, some assessment of glycosylation would be beneficial.*

We now demonstrate that select hexameric concatemers, both the highly active 6(1CT) and the inactive 6(6CT), expressed on the membrane surface are glycosylated (see shift in MW for samples treated with PNGase F in **Fig. R4**). We also find that dimeric and trimeric constructs are glycosylated. Data from these constructs are now presented in **Suppl. Fig. 2d & 3d** (with revised Methods, p.25).

Please note also that the biotinylation assays depicted in the original **Fig. 5b** demonstrate that all hexameric concatemers are present on the cell surface. In addition, the single channel analysis of concatenated PANX1 channels, which were activated by TEVp (or caspase) after the patch was excised from the cell, required their presence at the cell membrane. Thus, there is no doubt that these constructs are trafficked to the cell surface, and we now verify that they are also glycosylated.

5. *It is not clear to this reviewer what changes in the N-terminus (if any) of the PANX1 persists after TEVp cleavage? If the N-terminus is not truly native, could this also contribute to changes in channel conductances?*

We thank the Reviewer for pointing out that this was not clear.

We now provide the sequence variation for the non-native N-terminus that remained on the subunits following TEVp cleavage (or caspase cleavage) of the concatemeric channels (see **Fig. 2a** and p. 21). In brief, the N-terminus of the first subunit is unaltered, but the N-termini of the following subunits have a 3 amino acid extension from native channels (GLD). However, as

mentioned above (see point 2), the conductance properties of the fully cleaved concatemeric channels (i.e., after caspase) matched perfectly those of the monomeric wild type channel, suggesting that the addition of these sequences to the N-terminus of those following subunits was without noticeable effect on unitary conductance.

6. *The Na/K ATPase blot in Figure 5B is not very convincing and this ATPase is totaling missing in the 6(6CT) lane for some reason?*

The Na/K ATPase should not be present in the final 6(6CT) lane. That lane represents a control sample for the streptavidin pull down (i.e., that final sample was not treated with biotin, please see bar over figure). The ATPase is apparent in the second-to-last lane, also 6(6CT), from the corresponding biotin-treated sample.

7. *It is not clear when the authors use wildtype PANX1 if it is truly unaltered and simply full length PANX1 with no tags or alterations. Please clarify throughout the manuscript. If it carries modifications, I would argue that it is no longer wild type. Also, there are now isoforms of PANX1 that have been reported. Which isoform is being used and is it possible that differences in reported channel conductances in the literature could be linked to the different PANX1 isoforms?*

We again thank the Reviewer for pointing this out. Indeed, as suspected by the Reviewer, most of the recordings from “wild type” channels presented in the manuscript are actually from epitope-tagged constructs. To avoid this confusion, we now refer to these as “full length” channels, rather than “wild type”, and indicate which tags are present on the channels.

To address this point more directly, we have now also recorded from untagged recombinant full length PANX1 channels expressed in HEK293T (n=3 patches, see **Fig. R5**). As with the corresponding tagged versions (cf. **Fig. 1b-d**), these channels are silent before caspase treatment and

are activated by caspase-cleavage to produce an outwardly-rectifying channel (93.0 ± 4.8 pS and 11.4 ± 1.2 pS, at depolarized and hyperpolarized potentials). Thus, we find no effect of the epitope tag on channel properties; these new data are now presented in **Suppl. Fig. 1b**.

It is important to also mention that the epitope tags (GFP, FLAG) are always placed at the C-terminal end of the channel. Therefore,

they are removed by caspase cleavage at the C-terminus during channel activation. We also note that conductance properties of caspase-cleaved PANX1 channels in Jurkat T cells (i.e., native, untagged wild type channels) are comparable to those of recombinant, tagged versions of the channel (see **Fig. 6b**).

The human PANX1 has two isoforms. We used the reference isoform (“Isoform 1”, 426 amino acids), which is the full length sequence, whereas the Dahl laboratory used the alternative isoform (“Isoform 2”, 422 amino acids; see Bao *et al.*, 2004, *FEBS Lett.*), which is missing amino acids 401-404 (GMNI). However, the different conductance reported by two groups is unlikely to be explained by the use of different isoforms because the missing amino acids are located distal to the C-terminal caspase site (DVVD, residues 376–379) so that the difference in sequence between two isoforms is lost in PANX1 channels following caspase cleavage. It is beyond the

scope of the current study to test if elimination of the 4 C-terminal residues (GMNI) allows Isoform 2 of PANX1 to be activated by high [K]_e and attain the higher ~500 pS conductance. Nonetheless, we thank Reviewer for pointing this out as a potential explanation for discrepancy between different single channel properties of PANX1, and have added information about channel isoforms in the Methods (p. 20) and Discussion (p. 17).

8. *What does it mean that “All fits assume ~65% of GFP fluorescence is available for photobleaching”? How was this determined?*

We now clarify (see p. 25). In brief, some fraction of the GFP molecules is inevitably “pre-bleached or misfolded” before the experiment, and thus not available for photobleaching. This fraction was estimated empirically by use of the binomial equation, where the least-squares fitting includes a variable term for the GFP that remains available for photobleaching. For our analysis, we found that 65% of available GFP provided the best fit for the hexameric conformations (i.e., for a trimer of dimers, or a dimer of trimers). The statement referenced in the Reviewer comment indicates that we applied the same 65% value to fits when assuming channels could have alternative stoichiometries. Note that we also fitted the data to those various channel stoichiometries without constraining fractional GFP availability; with this additional analysis, the smallest deviation from the data was always seen with fits to the hexameric conformation (see **Suppl. Fig. 2a**).

9. *The authors must show some actual images of HEK293T cells that expressed the various PANX1 concatemers before and after TEVp co-expression? It is remarkable that these complex protein products are not mistargeted to some degree as the cell may recognize them as potentially misfolded protein complexes.*

We now provide the images requested (**Suppl. Fig. 3c; see Methods in p. 23**). It is beyond the scope of the study (or the interest of the authors) to track the fate of protein products from the concatemeric constructs. Based on our surface biotinylation assays, we can attest to the fact that the various concatemers (dimer, trimer, hexamer) all traffic to the cell membrane at roughly comparable levels to the monomeric subunits (relative to overall expression), where they clearly make functional channels. Indeed, we also find this remarkable (and powerful), and we hope this demonstration proves helpful to other groups studying similar channels.

10. *In the abstract and later in the manuscript the authors make reference to these findings possibly being applicable to connexins and other channel types. This should be removed as there is no evidence presented in the manuscript to support this notion or speculation. Specifically, the use of concatemeric connexins has been problematic and not particularly informative.*

As mentioned in point #9, we feel that this work may provide some impetus for other groups to attempt a similar approach. However, it is true that we have not established the utility of these concatemers in these other channel contexts, and will take the Reviewer’s advice to refrain from sounding too encouraging a note regarding this approach.

We also removed the indicated statement from the Abstract and deleted a point in the discussion where we had previously explicitly advocated use of concatemers. We prefer to retain our speculation at the conclusion of the paper that similar mechanisms might be relevant for related channels; however, in deference to this concern and so as not to mislead any unsuspecting readers, we state clearly that this is a speculation.

REVIEWERS' COMMENTS:

Reviewer #1 (Remarks to the Author):

The revised submission by Chiu et al describes a novel mechanism for activation of pannexin channels by c-terminal cleavage and $\alpha 1$ -receptor activation. The paper is logically presented, the rationale strong and the experiments well designed. Overall I found the data compelling and the discovery of this activation mechanism is likely to have impact in the pannexin field and beyond. I have a few minor points that require clarification or correction before I could recommended the paper for final acceptance.

1) With regards to appearance of the pronounced pore in the EM images following c-terminal cleavage: The author's point out that this appears to be restricted to one face of the channel (presumably intracellular). To my mind, it is this asymmetrical change that could account for the substantially lower conductance observed in their work compared to others. Perhaps other mechanisms of activating the channel also cause dilation of the (presumably) extracellular face of the pore. As such, the authors need to be more selective in their use of language. For example, the phrases on pg 5, line 121 and pg 8 line 198 should be modified to recognize the asymmetrical nature of the change in pore size. I also recommend that the authors add a sentence or two in the discussion that indicates other mechanisms of activation are possible, leading to different channel conductances and that a lack of apparent change in the (extracellular) face of the pore could restrict high conductance states.

2) One of the central conclusions is that the panx1 channel is not voltage-dependent because the Popen is not altered (Fig 1), rather there is a non-linear change in unitary conductance. This seems like a "splitting hairs" definition of voltage dependence because for the whole-cell, voltage-dependence is evident as outward rectification, although it may arise from changes in conductance not open probability. However, in Figs 3&4, there is an apparent change in Popen, which makes the claim that voltage-dependence arises from the change in conductance less clear. What happens to Popen for each concatamer versus voltage (i.e. what do the plots of Fig 4d look like at different voltages?) If Popen is truly voltage-independent different voltages should have similar linear relationships to the one shown for +80mV.

3) finally, similar to point 1 above, I would like the authors to be careful in their use of language. They need to avoid generalizing the $\alpha 1$ receptor activation as "receptor mediated mechanism". There is no evidence presented that the present G-protein coupled receptor mechanism will be applicable to all receptors that regulate the channel.

Reviewer #2 (Remarks to the Author):

Chiu et al. have responded to the reviewers by additional experiments and adaptations of the text. The paper is ready for publication in nature communications.

Reviewer #3 (Remarks to the Author):

The authors have adequately addressed the issues that I raised in the original submission. Nice study.

POINT-BY POINT RESPONSE TO REVIEWER COMMENTS

Reviewer 1

General Comments

The revised submission by Chiu et al describes a novel mechanism for activation of pannexin channels by c-terminal cleavage and α_1 -receptor activation. The paper is logically presented, the rationale strong and the experiments well designed. Overall I found the data compelling and the discovery of this activation mechanism is likely to have impact in the pannexin field and beyond. I have a few minor points that require clarification or correction before I could recommend the paper for final acceptance.

We appreciate these comments and have addressed the additional minor points below.

Main concerns:

1. *With regards to appearance of the pronounced pore in the EM images following c-terminal cleavage: The author's point out that this appears to be restricted to one face of the channel (presumably intracellular). To my mind, it is this asymmetrical change that could account for the substantially lower conductance observed in their work compared to others. Perhaps other mechanisms of activating the channel also cause dilation of the (presumably) extracellular face of the pore. As such, the authors need to be more selective in their use of language. For example, the phrases on pg 5, line 121 and pg 8 line 198 should be modified to recognize the asymmetrical nature of the change in pore size. I also recommend that the authors add a sentence or two in the discussion that indicates other mechanisms of activation are possible, leading to different channel conductances and that a lack of apparent change in the (extracellular) face of the pore could restrict high conductance states.*

We have now reinforced our statements in the discussion indicating that different mechanisms of channel activation are possible, and that properties associated with those alternatively activated PANX1 channels may be distinct (see p. 18, l. 401-403).

We describe in detail the asymmetrical nature of the “pore” diameter in the caspase cleaved PANX1 channel (see p. 4-5, l. 62-73). The EM data we provide for the TEVp-activated concatemeric channel shows only the presumed cytoplasmic view (see Fig. 3c), and therefore we make no claims regarding asymmetry to avoid overinterpretation of those data.

Given the uncertainties associated with negatively-stained EM images, especially those prepared from two different laboratories, we are not comfortable with attributing differences in single channel conductance between groups to variations in the appearance of the “pore” in these images (size, symmetry). We would nevertheless point out that the overall particle size is approximately the same in the datasets from the two studies, but the smallest “pore” diameter for the cleavage-activated PANX1 channel on the presumed extracellular side is actually slightly bigger than the largest “pore” diameter reported by the Dahl and Sosinsky groups in high extracellular K^+ (i.e., 58 Å vs. 54 Å, Wang et al., 2014); the intracellular “pore” diameter we find is even larger still (100 Å). The qualitative nature of these images notwithstanding, we do not think the smaller conductance reported here for a cleavage-activated channel is easily explained by differences in symmetry or apparent “pore” size (which is actually larger than that for K^+ -exposed channels).

2. One of the central conclusions is that the panx1 channel is not voltage-dependent because the Popen is not altered (Fig 1), rather there is a non-linear change in unitary conductance. This seems like a "splitting hairs" definition of voltage dependence because for the whole-cell, voltage-dependence is evident as outward rectification, although it may arise from changes in conductance not open probability.

When considered with respect to the voltage dependence of whole cell current, the Reviewer likens our conclusions regarding contributions of open channel rectification in the absence of voltage-dependent gating to "splitting hairs". However, we must respectfully disagree: Defining these properties is fundamental for a mechanistic understanding of channel function.

Our data show prominent outward rectification from cleavage-activated PANX1 channels, with no difference in open probability (P_o) over a wide range of membrane potentials; channels activated by α_1 -adrenoceptors show similar strong outward rectification. If the outward rectification of whole cell current was due to voltage-dependent "gating", then an increase in P_o would have been observed at depolarized membrane potentials, which would imply the existence of a "sensor" capable of translating the potential difference across the membrane bilayer to an effect on the channel "gate". Because this was not the case, there is no need to invoke the presence of a voltage sensor to activate cleaved PANX1 channels. Rather, the data indicate that whole cell rectification reflects open channel rectification, and imply that there must be some biophysical/structural basis for the observed differences in resistance to outward vs. inward current.

However, in Figs 3&4, there is an apparent change in Popen, which makes the claim that voltage-dependence arises from the change in conductance less clear. What happens to Popen for each concatamer versus voltage (i.e. what do the plots of Fig 4d look like at different voltages?) If Popen is truly voltage-independent different voltages should have similar linear relationships to the one shown for +80mV.

Unfortunately, there appears to be a misunderstanding of the P_o data presented in Figs. 3 & 4; these differences in P_o are related to the number of intact C-termini and do not provide any information on voltage dependence of the channels (i.e., the records in Fig. 3 are all taken at the same voltage; and the X-axis in Fig. 4d is the number of C-tails and not membrane voltage). A similar linear relationship at a different but constant voltage, as suggested by the Referee, would likewise reveal effects of C-tail number on P_o without addressing voltage-dependent gating.

Although we show that gating is unaffected by membrane voltage in the fully cleavage-activated channel (see Fig. 1e), it is formally possible that P_o for partially cleaved channels might show some voltage-dependent gating (i.e., they could behave unlike the fully activated channel). Unfortunately, in channels with more intact C-tails, the diminutive single channel currents and retained open channel rectification make measurements of P_o less reliable, especially at the negative membrane potentials required for determining voltage dependence. Even so, such a biphasic effect of C-tail removal on gating would be surprising indeed. It would require that voltage-dependent gating was induced by removal of a few C-tails and then subsequently lost as additional C-tails were deleted. Moreover, we note that all concatameric constructs show single channel rectification, and all display whole cell I - V properties that are identical to the fully cleaved PANX1 channel (*Inset*, Fig. 5a). Thus, it would also be a remarkable coincidence if identical I - V s among these constructs could arise from a gating process that first emerged and then disappeared during C-tail removal, and that those biphasic effects were precisely offset by concomitant (and undetected) differences in single channel rectification.

In short, the analysis suggested by the Reviewer will not address the issue of voltage-dependent gating, which we have already presented for the fully cleaved channel (see Fig. 1e). Furthermore, our data suggest that the possibility of some hidden voltage dependent gating in partially activated channels is highly unlikely. Nonetheless, in keeping with the general recommendation that additional qualifications of our results be provided (see point #1), we note explicitly in the re-

vised manuscript that the present observations regarding conductance and voltage-dependent gating pertain specifically to PANX1 channels that have been activated by either C-terminal cleavage or by α_1 -adrenoceptors (p. 18, l. 402).

3. *Finally, similar to point 1 above, I would like the authors to be careful in their use of language. They need to avoid generalizing the α_1 receptor activation as "receptor mediated mechanism". There is no evidence presented that the present G-protein coupled receptor mechanism will be applicable to all receptors that regulate the channel.*

We have now checked our language extensively to ensure that we state the described effects are observed with α_1 -adrenoceptor activation of PANX1 (see red text throughout). We also mention that similar effects are seen with channel activation by other G α_q -linked receptors (p. 14, l. 304). However, we acknowledge that channel properties may be different for other mechanisms of channel activation or in different cell contexts (p. 18, l. 402).

Reviewer #2 (Remarks to the Author):

Chiu et al. have responded to the reviewers by additional experiments and adaptations of the text. The paper is ready for publication in nature communications.

Reviewer #3 (Remarks to the Author):

The authors have adequately addressed the issues that I raised in the original submission. Nice study.